# K-th Nearest Neighbor (KNN) Entropy Estimates of Complexity and Integration from Ongoing and Stimulus-Evoked Electroencephalographic (EEG) Recordings of the Human Brain

**DOI:** 10.3390/e21010061

**Published:** 2019-01-13

**Authors:** Logan T. Trujillo

**Affiliations:** Department of Psychology, Texas State University; San Marcos, TX 78666, USA; logant@txstate.edu; Tel.: +01-512-245-3623

**Keywords:** Electroencephalography (EEG), EEG complexity, EEG integration, induced EEG, evoked EEG, resting state EEG, brain criticality, visual categorization

## Abstract

Information-theoretic measures for quantifying multivariate statistical dependence have proven useful for the study of the unity and diversity of the human brain. Two such measures–integration, *I(X)*, and interaction complexity, *C_I_(X)*–have been previously applied to electroencephalographic (EEG) signals recorded during ongoing wakeful brain states. Here, *I(X)* and *C_I_(X)* were computed for empirical and simulated visually-elicited alpha-range (8–13 Hz) EEG signals. Integration and complexity of evoked (stimulus-locked) and induced (non-stimulus-locked) EEG responses were assessed using nonparametric *k-th* nearest neighbor (KNN) entropy estimation, which is robust to the nonstationarity of stimulus-elicited EEG signals. KNN-based *I(X)* and *C_I_(X)* were also computed for the alpha-range EEG of ongoing wakeful brain states. *I(X)* and *C_I_(X)* patterns differentiated between induced and evoked EEG signals and replicated previous wakeful EEG findings obtained using Gaussian-based entropy estimators. Absolute levels of *I(X)* and *C_I_(X)* were related to absolute levels of alpha-range EEG power and phase synchronization, but stimulus-related changes in the information-theoretic and other EEG properties were independent. These findings support the hypothesis that visual perception and ongoing wakeful mental states emerge from complex, dynamical interaction among segregated and integrated brain networks operating near an optimal balance between order and disorder.

## 1. Introduction

### 1.1. Information-Theoretic Measures of Brain Integration and Complexity

It is well-established that neurocognitive functioning is tied to the dynamical interaction of neuronal networks in the brain. Of particular interest is the observation that the brain organizes itself into diverse local networks implementing specialized information processing (a phenomenon termed segregation) and global networks that combine the local networks into a unified information process distributed across the brain (a phenomenon termed integration) [1,2,3,4,5,6,7]. Moreover, the dynamic interplay of segregation and integration is highly complex, reflecting coordinated interactions among neural hierarchies that are neither fully regular nor fully random [4]. Multiple analytical metrics are available to estimate segregation, integration, and complexity in the brain [8,9,10] by directly characterizing “deviations from statistical independence among components of a neural system” [6] (p. 5033). Information-theoretic methods for quantifying statistical dependence among multiple variables have proven especially useful for this purpose; the application of such techniques to scalp-recorded electroencephalographic (EEG) measures of bioelectric brain activity is the topic of the study reported here.

This paper illustrates the application of two information-theoretic measures to a set, *X*, of EEG signals: The Tononi–Edelman–Sporns integration, *I(X)*, and interaction complexity, *C_I_(X)* [4,6]:(1)$$I(X)=\sum_{i=1}^{N_{s}}H(X_{i})-H(X),$$
(2)$$C_{I}(X)=H(X)-\sum_{i=1}^{Ns}H(X_{i}|X-X_{i})$$

*I(X)* and *C_I_(X)* are formed from the combination of different entropies, *H*, that quantify the uncertainty of the EEG signals and thus how informative they are. In Equations (1) and (2), *H(X_i_)* is the entropy of an individual EEG channel, *X_i_*, *H(X)* is the joint entropy of the coincident patterns, *X*, across all EEG signals, *N_s_*, and:(3)$$H(X_{i}|X-X_{i})=H(X)-H(X-X_{i})$$
is the conditional entropy of a single EEG signal, *X_i_*, given the state of the other EEG signals, *X–Xi*. *I(X)* indexes the overall deviation from statistical independence of the individual elements in a system, whereas *C_I_(X)* is a statistical measure of a system’s information content that results from the interactions among the system’s elements. There is a non-monotonic “inverted-U” relationship between complexity and integration [6,11]; see Figure 1. When *I(X)* = 0, all EEG channels are statistically independent and *C_I_(X)* is low (point 1 of Figure 1). When *I(X)* is maximal, all EEG channels are statistically dependent and *C_I_(X)* is also low (point 3 of Figure 1). However, when *I(X)* attains an intermediate value, then there is a heterogenous statistical dependence among the EEG signals and *C_I_(X)* attains a maximum value (point 2 of Figure 1). In the latter case, the EEG signals are neither completely independent nor completely dependent, which presumably reflects a highly integrated yet specialized state of the brain [4].

### 1.2. Ongoing Versus Stimulus-Elicited EEG Integration and Complexity

Several studies have applied information-theoretic metrics of integration and/or complexity to ongoing scalp-recorded electromagnetic signals [11,12,13,14,15,16,17]. The EEG signals recorded in these studies were “ongoing” in the sense that they reflected relatively continuous, steady-state neurocognitive processes elicited by a variety of mental or psychomotor activities (e.g., mental arithmetic, working memory, motor grasping, resting state, vigilant attention) that are not precisely time-locked to an external stimulus. However, many neurocognitive processes of interest arise from direct sensory stimulation that produce EEG signals reflecting a mixture of ongoing brain activity and brain activity elicited in response to external events. To date, there are few published studies quantifying *I(X)* and *C_I_(X)* for stimulus-elicited EEG states [12,18]. Moreover, none of these studies accounted for two methodological problems that must be addressed when applying measures of statistical dependence to stimulus-elicited EEG signals.

The first methodological problem is distinguishing between the contributions to the EEG entropy from EEG signals that are either stimulus-locked or not stimulus-locked to the onset of an eliciting environmental event (typically sensory stimulation) [19]. Non-stimulus-locked EEG signals (termed induced EEG activity) reflect unreliable background brain activity not related to stimulation per se and neurocognitive processes that are reliably elicited in response to an event, but with a variable onset, phase, and time course relative to event-onset across trials. Stimulus-locked EEG signals (evoked EEG activity) reflect neurocognitive processes that are reliably elicited by an event with a relatively consistent onset, phase, and time course relative to event-onset across trials. These two different contributions to the EEG entropy were distinguished in the present study by using a previously established method [20] to separate EEG signals into induced and evoked components before computation of *I(X)* and *C_I_(X)* (see Section 2.5, below).

### 1.3. K-th Nearest Neighbor Entropy Estimation for EEG Integration and Complexity

The second methodological problem that must be addressed when applying information-theoretic measures of statistical dependence to EEG signals is accounting for the latter’s statistical nonstationarity. The nonstationarity of EEG signals is a consequence of the fact that many neurocognitive processes involve the continuous activation and deactivation of a variety of neural sources that realize information-processing throughout the brain [21,22]. EEG signals are well-known to be nonstationary over long time intervals, but approximately mean- and trend-stationary over short-time periods [18,22,23]. However, EEG signals remain variance-nonstationary over time periods short enough to encompass the early stages of stimulus-elicited processes (~1 s or less) [22]. To the present author’s knowledge, all current *H* estimators assume that the data to which they are applied are stationary. This raises difficulties in their applications to EEG data in that such nonstationarities can distort entropy estimates to an extent depending on the nature of the nonstationarity and the choice of entropy estimator [24,25].

Fortunately, certain entropy estimators are minimally affected by the nonstationarity of a signal. One such estimator is the nonparametric k-th nearest neighbor (KNN) entropy estimator [26,27,28,29,30], which was used here to compute the entropies composing *I(X)* and *C_I_(X)*. Here, the entropy of a data space is estimated by approximating the multivariate probability density function, *f_X_(X)*, at each point, *x_i_*, in the space in terms of the ratio of the fraction of other points in a data point’s neighborhood to the volume of its neighborhood [29]:(4)$$H(X)=-E\left[\log f_{X}(X)\right]\approx -\frac{1}{N}\sum_{i=1}^{N_{p}}\log({\hat{f}}_{X}(x_{i})).$$
(5)$${\hat{f}}_{X}(x_{i})=\frac{k(x_{i})/N_{p}}{Vol_{i}}.$$
where *N_p_* is the number of data points in the space, *Vol_i_* is the volume of the neighborhood, and *k(x_i_)* is the number of points in the neighborhood of *x_i_* other than *x_i_*.

KNN estimators can reliably detect changes in the dynamical complexity of variance-nonstationary data at the cost of introducing a small bias [24]. However, the impact of such bias is likely reduced when using this estimator to compute *I(X)* and *C_I_(X)* because these quantities are defined in terms of entropy differences, which should tend to cancel out such a bias in the subtraction. Moreover, KNN entropy estimators are nonparametric and thus avoid entropy distortions that may arise when using an estimator derived via specific distributional assumptions of the data that are inaccurate, as was recently observed when a Gaussian entropy estimator was used to compute the complexity of non-Gaussian EEG data [11]. Finally, KNN entropy estimators are easy to implement and efficient to compute for multidimensional variable spaces [26,29], such as a set of EEG signals (although there are still limits to this efficiency for large spaces; see Section 4.3, below).

In the present study, a geometric KNN estimator (G-KNN) [29] was employed that uses geometrical volume elements computed via singular value decomposition (SVD) of the neighborhood of each data point to accurately model irregularities in the local geometry of a data space:(6)$$\begin{array}{l} H_{G-KNN}(X)=\log(N_{p})+\log(\pi^{d/2}/\Gamma (1+d/2))-\frac{1}{N_{p}}\sum_{i=1}^{N_{p}}\log(k(x_{i}))\\ \;+\frac{d}{N_{p}}\sum_{i=1}^{N_{p}}\log(\varepsilon (x_{i},k))+\frac{1}{N_{p}}\sum_{i=1}^{N_{p}}\sum_{l=1}^{d}\log(\frac{\sigma_{i}^{l}}{\sigma_{i}^{1}}) \end{array}$$
where *N_p_* is the number of data points in the space, *d* is the dimension of the data space, *k(x_i_)* is the number of neighborhood points within the volume of an ellipse centered on that point, *ε(x_i_,k)* are the Euclidean *k-th* nearest neighbor distances, and *σ^l^_i_* are the singular values for each point, *x_i_*. The specific procedure for computing *K_G-KNN_(X)* may be found in [29]. Note that in the case of *d* = 1, log(σ^l^_*i*_/σ^1^_*i*_) = log(σ^1>^_*i*_/σ^1^_*i*_) = 0, and Equation (6) reduces to a variant of the Kraskov-Stögbauer-Grassberger (KSG) KNN entropy estimator [26,28]:(7)$$H_{KSG}(X)=\log(N_{p})+\log(\pi^{d/2}/\Gamma (1+d/2))-\frac{1}{N_{p}}\sum_{i=1}^{N_{p}}\log(k(x_{i}))+\frac{d}{N_{p}}\sum_{i=1}^{N_{p}}\log(\varepsilon (x_{i},k)).$$

For the computation of *I(X)* and *C_I_(X)* in the present study, Equation (6) was used to compute the joint entropy, *H(X)*, and conditional entropy, *H(X_i_|X–X_i_)*, whereas Equation (7) was used to compute the individual channel entropies, *H(X_i_)*.

### 1.4. The Present Study

The present study utilized KNN-based entropy estimation to compute *I(X)* and *C_I_(X)* for alpha-range (8–13 Hz) human EEG signals acquired during a task involving the visual perception and categorization of simple visual stimuli (Gabor stimuli; see Section 2.3, below). The present analysis was restricted to alpha-band-filtered signals for two reasons. First, such filtering removes data characteristics (e.g., trends, spikes) that reflect nonstationarities other than variance-nonstationarity, a fact consistent with the suggestion that EEG integration and complexity computations will be more accurate when performed on data within a narrow frequency range [16]. Second, there is evidence that visual-evoked EEG signals have significant energy in the alpha range that is related to visual perception [19,31,32].

The main goal of the present study was to compare any pre- versus post-stimulus changes in *I(X)* and *C_I_(X)* that may occur as a result of a change from ongoing to induced and evoked stimulus-elicited states of the brain. In addition, the effects of task difficulty and complexity were examined by having participants perform two versions of the visual task that differed in terms of the complexity of the category structure and the number of stimulus exemplars per category (see Section 2.3, below). *I(X)* and *C_I_(X)* were computed separately for induced and evoked EEG responses. Spectral power and interelectrode synchronization of the induced and evoked EEG responses [10,33] were also computed in order to assess how these signal properties changed across different states of EEG complexity and integration (see Section 2.7, below). Furthermore, EEG dipole source modeling [34,35,36] was performed in order to assist the interpretation of the observed integration and complexity measures given that the latter are confounded by the presence of spurious interdependencies arising from the spatial blurring of scalp-recorded cortical signals due to volume conduction through the head [11] (see Section 2.9, below). These analyses demonstrated a sensitivity of KNN-based EEG integration and complexity measures to neural differences between ongoing and stimulus-evoked EEG states during visual perception and categorization. However, the meaningful interpretation of these measures in terms of integration and segregation of underlying cortical activity required the additional assessment of EEG spectral power and synchronization properties.

The secondary goal of the present study was to use KNN-based estimation to compute alpha-range EEG *I(X)* and *C_I_(X)* during an ongoing wakeful resting state of the brain (interleaved counterbalanced eyes-open and eyes-closed wakeful states; see Section 2.2, below). These wakeful states are characterized only by induced EEG rhythms because ongoing brain activity is not time-locked to external events [20]. The recording and analysis of resting state EEG was performed in order to compare the difference between task-relevant and restful induced EEG integration and complexity, and in order to compare the outcome of the KNN-based estimation of wakeful resting state integration/complexity to past studies that computed alpha-range resting state *I(X)* and *C_I_(X)* using Gaussian-based entropy estimators [9,11,16,17]. It was found that the KNN-based measures replicated *I(X)* and *C_I_(X)* patterns previously observed during ongoing wakeful EEG states obtained via Gaussian entropy estimation. This supports the replicability of the observed pattern of wakeful resting state EEG integration and complexity, as well as the validity of KNN entropy estimation for the computation of EEG integration and complexity.

## 2. Materials and Methods

### 2.1. Participants

Sixteen Texas State University undergraduates participated for course credit or monetary payment (8 female, 8 male, mean age = 21.8 ± 8 years, age range = 18–26). All participants gave written informed consent in accordance with the Declaration of Helsinki. The Institutional Review Board at Texas State University approved this study.

### 2.2. Resting State EEG

After consent, participants underwent setup for EEG recording. During the setup, participants completed several questionnaires indexing demographic and health information, sleep quality and quantity, emotion and mood states, current sleepiness and attentional states, and task-related mental workload. However, the results of these questionnaires are not relevant to the topic of this paper and are not reported here.

After completion of the EEG setup, participants underwent 8 min of resting state EEG recording while sitting quietly in a comfortable padded chair in a darkened room (4 min eyes open and 4 min eyes closed interleaved in 1-min intervals; eyes open/closed order was balanced across participants). Participants were instructed to remain relaxed and alert during recording.

### 2.3. Categorization Task

Following recording of the resting state EEG, participants then performed two simple visual categorization tasks that differed in terms of difficulty, the complexity of the stimulus category space, and the number of exemplars per visual category (1-Exemplar Task, 2-Exemplar Task; Figure 2). Visual stimuli were circular sine-wave gratings (Gabor stimuli) organized into two categories on the basis of specific combinations of spatial frequency (number of bars and bar width per patch) and orientation (tilt angle of lines) of the gratings. Sine-wave gratings spanned ~4.2° of the visual angle at a viewing distance of 75 cm.

In the 1-Exemplar Task, there was only one exemplar per category (i.e., a total of two unique stimuli were presented; Figure 2a) so that each category was defined by the individual identity of each stimulus in terms of their specific, distinguishing spatial frequencies and orientations. This simple category structure was easily learned by participants (see Section 3.3, below). In the 2-Exemplar Task (Figure 2b), each category consisted of two exemplars (4 stimuli total) that possessed opposing feature characteristics. For example, Category 1 stimuli may have possessed low spatial frequency Gabor bands at acute angles less than 45° or high spatial frequency bands at acute angles greater than 45°, whereas Category 2 may have possessed low spatial frequency Gabor bands at acute angles greater than 45° or high spatial frequency bands at acute angles less than 45°. This produced a complex visual category structure that was more difficult for participants to learn (see Section 3.3, below).

Participants were assigned to one of eight different versions of the 1-Exemplar Task and one of four versions of the 2-Exemplar Task, where each version differed in terms of the particular combinations of spatial frequency and orientation that defined the categories (see Appendix A for a list of these definitions). The across-participant counterbalancing of the different task versions allowed the stimuli for each category to be approximately matched in terms of mean luminance, contrast, and spatial frequency across participants in the 1-Exemplar Task and matched for each participant in the 2-Exemplar Task. Stimulus spatial frequencies ranged from ~1.6 to 4.0 cycles per degree of visual angle across the different versions of the two categorization tasks, whereas stimulus orientations ranged from ~18 to 72 angular degrees from the vertical.

Figure 2c displays a typical categorization task trial. A fixation cross was presented for a variable interstimulus interval (ITI; 1200–1500 ms) at the center of a computer screen, followed by the to-be-categorized-stimulus for 1000 ms. The stimulus was then removed and the participant was visually queried about the stimulus category. The subject had a maximum of 2000 ms to indicate a categorization decision by pressing one of two buttons on a computer mouse. This was followed by visual feedback (“Correct”, “Incorrect”, or “No Response”) for 500 ms before a new trial begun. Participants categorized 400 sine-wave gratings presented across five blocks for each task session (80 stimuli per block, 40 stimuli per category per block). Prior to task performance, participants were familiarized with the procedures of each task and were shown all the stimuli, with explicit information about which stimuli were to be assigned to each category. The order of the 1-Exemplar and 2-Exemplar tasks was balanced across participants.

### 2.4. EEG Recording and Preprocessing

Seventy-two EEG channels (Figure 3) were recorded from the scalp of each participant during resting and categorization task performance (BioSemi Active II system; 24-bit DC mode; 2048 Hz initial sampling rate downsampled online to 256 Hz; common mode sense reference electrode located between sites PO3 and POZ) and preprocessed using standard procedures [11,37]. EEG preprocessing included artifact-scoring, bad channel interpolation, and bandpass filtering (8–13 Hz; 8449 point zero-phase shift FIR filter; 0.1 Hz transition bands). Bad channel interpolation was implemented via a spherical spline interpolation algorithm (m = 5; 50-term expansion) [38] applied to the other channels; the mean number of interpolated channels was 2.8 ± 0.9 (less than 4% of all channels). EEG data were also transformed into Laplacian-based current source densities in order to reduce the effects of volume-conduction on the integration and complexity measures [11,39] (see also Section 2.8.2, below). Resting EEG data were divided into 1 s epochs with 50% overlap; on average, 340 ± 16 eyes closed and 299 ± 17 eyes open trials were retained after artifact rejection.

Categorization task EEG data were divided into two classes of epochs, pre-stimulus epochs ranging from −1000 ms–0 ms relative to stimulus onset, and post-stimulus epochs ranging from 0 ms–1000 ms relative to stimulus onset. Categorization task trials with RTs < 200 ms were excluded from further analysis. As specific differences between categorization accuracy were not of interest in this study, EEG trials were aggregated across correct and incorrect categorizations to increase the signal-to-noise ratio and statistical power. After artifact rejection, 219 ± 18 pre-stimulus trials and 321 ± 15 post-stimulus trials remained on average for the 1-Exemplar Task and 214 ± 25 pre-stimulus trials and 345 ± 10 post-stimulus trials remained on average for the 2-Exemplar Task.

All epochs were then adjusted by subtracting the mean signal across an epoch for each electrode separately. Finally, in order to reduce the computation time of entropy estimates to a tractable level (see Section 4.3 for further discussion), EEG epochs were downsampled by half in time (from 256 Hz to 128 Hz) and space (from 72 to 36 EEG signals). Temporal downsampling was accomplished via the MATLAB downsample function, whereas spatial downsampling was accomplished by averaging signals from neighboring electrode pairs across the scalp (see Figure 3). Preprocessing analysis was also implemented using the EEGLAB toolbox [40] for the MATLAB computing software environment (The MathWorks, Inc., Natick, MA, USA).

### 2.5. Estimation of Induced and Evoked Categorization Task EEG Signals

Induced and evoked EEG signals present during the categorization task were estimated using a previously established method [20]. First, a stimulus-locked evoked response was estimated for each time point by averaging the *i* = 1:*N_s_* alpha-range bandpass-filtered EEG signals across *j* = 1:*N_tr_* trials:(8)$$X_{i}{\left(t\right)}_{Evoked}=\frac{1}{N_{tr}}\sum_{j=1}^{N_{tr}}X_{ij}(t).$$

Then, the evoked EEG response (also called an event-related potential or ERP) was subtracted from each trial of the alpha-range bandpass-filtered EEG to create a measure of the induced response for each electrode and trial:(9)$$X_{ij}{\left(t\right)}_{Induced}=X_{ij}(t)-X_{i}{\left(t\right)}_{Evoked}.$$

Induced and evoked EEG responses were computed for each participant, categorization task, and condition.

In contrast to the categorization task, wakeful resting state EEG signals were not decomposed in this manner because such signals do not contain stimulus-locked components and thus only reflect induced brain states [20].

### 2.6. Computation of EEG Integration and Complexity

Integration and complexity were computed for each induced EEG response epoch and averaged evoked EEG response according to Equations (1) and (2), where the individual channel entropies, *H(X_i_)*, were computed using the KSG entropy estimator (Equation (7)) and the joint entropy, *H(X)*, and conditional entropies, *H(X_i_|X-X_i_)*, were computed using the G-KNN entropy estimator (Equation (6)). *I(X)* and *C_I_(X)* were computed for the EEG signals by estimating *H* from data pooled across a time interval of interest. This is equivalent to treating each set of EEG signals as a jointly-distributed space of *N* = 36 variables, with each time point as an individual observation drawn from the joint-distribution. *I(X)* and *C_I_(X)* were computed from the time course of the trial-averaged evoked EEG signal, and from the time courses of each trial of the induced EEG signals before averaging the induced EEG integration and complexity across trials to produce a final estimate. EEG integration and complexity were computed for each participant and data condition (resting data: Eyes open, eyes closed; categorization tasks: Pre- and post-stimulus intervals). All integration and complexity values were computed in units of bits.

Although the value of the nearest neighbor parameter, *k*, should be small to provide a local estimate of the geometry of the joint variable data space, here, the value of *k =* 36 was chosen. This value was chosen because, in principle, *k* for the G-KNN estimator must at least equal the number of axes of the local ellipsoidal neighborhood computed around each data point by SVD (see Section 1.3, above), which in turn is equal to the dimension of the joint variable space of the data under analysis [29]. Hence, the chosen value of *k* was a compromise that balanced the latter criterion with the need for locality of the estimate. Although the KSG estimator for *H(X_i_)* is only subject to the locality criterion and the G-KNN estimator for *H(X_i_|X-X_i_)* operates on a space with a number of dimensions reduced by 1 relative to the full joint variable space, the value of *k =* 36 was used for all entropy estimates in order to avoid potential scaling differences that might arise from the use of different values of *k* across the different estimations. The use of a larger value of *k* than necessary for *H(X_i_)* and *H(X_i_|X-X_i_)* may have slightly reduced the entropy estimation accuracy, but this is likely mitigated by the fact that larger values of *k* also reduce the variance of KNN entropy estimates as well [41,42].

### 2.7. Computation of EEG Power and Phase Synchronization

#### 2.7.1. EEG Power

Spectral power and synchronization of the induced and evoked EEG responses were also assessed in order to facilitate the interpretation of EEG integration and complexity in terms of the energetic and temporal properties of the EEG signals. Spectral power was computed using a simple procedure appropriate for band-filtered signals [20,43] in which the values of the filtered EEG signal amplitudes (in units of *µV*) are squared and additively combined to yield trial-averaged EEG power values (in units of *µV*^2^) as follows:(10)$$P_{i}{\left(t\right)}_{Evoked}=X_{i}{\left(t\right)}_{Evoked}^{2},$$
(11)$$P_{i}{\left(t\right)}_{Induced}=\frac{1}{N_{Tr}}\sum_{j=1}^{N_{tr}}X_{ij}{\left(t\right)}_{Induced}^{2}.$$

Spectral power was then converted to global field power (GFP) by aggregating across electrodes as follows [44]:(12)$$GFP{\left(t\right)}_{Evoked}=\sqrt{\sum_{i=1}^{N_{s}}P_{i}{\left(t\right)}_{Evoked}}$$
(13)$$GFP{\left(t\right)}_{Induced}=\sqrt{\sum_{i=1}^{N_{s}}P_{i}{\left(t\right)}_{Induced}}$$

Note that *GFP* is expressed in units of *µV*. Induced and evoked global field power were then averaged across time to yield aggregate *GFP* measures for each participant and categorization task condition. As the resting state EEG did not contain stimulus-locked components, induced *GFP* was computed for these signals by applying Equations (11) and (13) to the original, non-decomposed, preprocessed resting state EEG data (see Section 2.5, above).

#### 2.7.2. EEG Synchronization

EEG synchronization was indexed by estimating an order parameter, *Λ(t)*, that characterizes the synchronization (coupling) of a population of oscillators [10]. First, an analytic signal was calculated for each EEG signal on a given trial via Hilbert Transformation:(14)$$A_{i}(t)=(X_{i}(t)-<X_{i}(t){>}_{t})+\sqrt{-1}\cdot H_{i}(t),$$
where:(15)$$H_{i}(t)=\frac{1}{\pi }p.v.\int_{-\infty }^{\infty }\frac{X_{i}(\tau )}{t-\tau }\;d\tau .$$

The order parameter, *Λ(t)*, is then given as:(16)$$\Lambda (t)=\frac{\sum_{i=1}^{N_{s}}A_{i}(t)}{\sum_{i=1}^{N_{s}}|A_{i}(t)|},$$

High values of *Λ(t)* indicate a strong synchronization between EEG signals on each trial, whereas low values indicate weak synchronization. For the induced EEG response, the order parameter, *Λ(t)_Induced_*, was computed for each individual trial and then averaged across time and trials to yield an aggregate measure of synchronization for each participant and categorization task condition. For the evoked EEG response, the order parameter, *Λ(t)_Evoked_*, was computed from the trial-averaged EEG signals and then averaged across time for each participant and categorization task condition. Only *Λ(t)_Induced_* was computed for the wakeful resting state EEG signals.

### 2.8. Statistical Analysis of Observed EEG Measures

Two kinds of statistical analysis of the EEG data were performed, a parametric approach that assessed potential differences between EEG data conditions (under the assumption that these differences were normally-distributed) and a non-parametric surrogate data testing approach that assessed the degree to which EEG complexity and integration may arise from random or spuriously-coincident activity among the EEG signals.

#### 2.8.1. Parametric Statistics

All parametric statistical analyses reported in this paper were performed using the SPSS software package (IBM Corporation, Armonk, NY, USA). Analysis of the across-participant statistical distributions of the EEG property measures (spectral power, synchronization, complexity, and integration) showed that their distributions did not significantly deviate from normality and thus met parametric assumptions [45,46,47]; see Appendix A. Categorization task EEG properties were analyzed via 2 x 2 repeated-measures ANOVA with factors of Task (1-Exemplar, 2-Exemplar) and Time Interval (Pre-stimulus, Post-stimulus). Induced and evoked EEG properties were compared by averaging categorization task EEG properties across the factor of Task and then performing 2 x 2 repeated-measures ANOVA with factors of EEG Type (Induced, Evoked) and Time Interval. Resting state EEG properties were analyzed via one-way repeated-measures ANOVA with a factor of Resting State Condition (Eyes Closed, Eyes Open). In order to contrast EEG brain states during categorization and wakeful rest, categorization task EEG properties were averaged across the Task factor and then compared to the resting state EEG properties via one-way repeated-measures ANOVA with a factor of Data Condition (Prestimulus, Poststimulus, Eyes Closed, Eyes Open). Given that the Data Condition factor involved more than two levels, the *p*-values of these omnibus tests were adjusted using the Greenhouse–Geisser correction for nonsphericity [48]. For ease of interpretation, reports of all *F* tests subject to Greenhouse-Geisser correction include uncorrected degrees of freedom, corrected *p*-values, and the Greenhouse-Geisser epsilon value, *ε*. Post-hoc comparisons for all tests were corrected for multiple comparisons using the Holm-Bonferroni procedure [49]; corrected *p*-values are indicated in the text.

In addition, regression analyses relating EEG power and synchronization to *I(X)* and *C_I_(X)* were conducted using generalized estimating equations (GEEs) [50,51], a generalized multivariate regression procedure that accounts for correlations across repeated measure levels while estimating robust and unbiased parameter standard errors. The GEE analysis assumed a normal distribution with an identity link, a robust covariance estimate, a maximum likelihood-estimate scale parameter, and an exchangeable correlation matrix. Two regression analyses were performed for each set of induced and evoked EEG properties separately. The first analysis collapsed across the two categorization tasks and time intervals in order to assess the relationship between absolute levels of alpha-band EEG power/synchronization and integration/complexity. The second analysis was performed after computing post- minus prestimulus differences for each EEG property in order to compare the relationship between the stimulus-related changes in EEG power/synchronization and EEG integration/complexity. Regressions were performed with EEG *I(X)* and *C_I_(X)* as dependent variables and the corresponding induced or evoked EEG power/synchronization as independent covariates. The reports of GEE analyses in this paper include standardized regression coefficients and tests of model effects (Wald χ^2^ statistic values, associated degrees of freedom, *p*-values).

Finally, analysis of the across-participant statistical distributions of participant behavior (accuracy, reaction time) showed that the reaction time data of both categorization tasks and the accuracy data of the 2-Exemplar task each met parametric assumptions, but the accuracy data for the 1-Exemplar task did not (see Appendix A). Thus, across-task comparisons of participant reaction time were performed via one-way repeated measures ANOVA with a factor of Task, whereas the comparison of participant accuracy was performed using a non-parametric Wilcoxon Signed Rank Test. In addition, GEE-based regression analyses were performed relating behavior to the poststimulus-prestimulus differences in *I(X)* and *C_I_(X)* difference scores to quantify the relationship between stimulus-related changes in EEG integration/complexity and task behavior. Regressions were performed separately for each behavioral measure. For regressions involving reaction time and 2-Exemplar task accuracy, the behavior measures were treated as the dependent variables and EEG integration and complexity as independent covariates. For regressions involving 1-Exemplar task accuracy, EEG integration and complexity were treated as the dependent variable and accuracy as an independent covariate. This mitigated the non-normality of the 1-Exemplar task accuracy data; dependent variables are model-dependent in GEE analyses, whereas independent variables are not [52]. Also, the GEE analyses used a robust covariance estimator, which allowed for a model-free estimate of the data covariance structure [53]. For all regression analyses of behavior, Task was included as an independent within-participant factor in order to assess observed task-related behavioral differences (see Section 3.3, below).

#### 2.8.2. Surrogate Statistics

Interpretation of the interdependence of scalp-recorded EEG signals is often confounded by the dispersion of these bioelectric signals as they travel through the head from the cortex to the scalp. This volume conduction may produce apparent statistical dependencies among EEG sensors when the originating brain activity involves spuriously-coincident, but otherwise independent, neural sources [54,55,56]. Here, a surrogate data technique was used to statistically assess the level of spurious dependencies present in the scalp-EEG signals; the method used here was a modification of that used by [11,57,58]. This method involved the creation of surrogate EEG data from a superposition of statistically independent sources with randomly-shifted EEG signal phases. Such surrogate data has similar energetic and temporal characteristics as the observed data, but with complexity and integration arising from volume conduction alone.

The first step was to decompose the time and space reduced observed EEG data (see Section 2.5, above) into independent signals via independent components analysis (ICA) [59]. Second, any remaining statistical dependencies among the ICA components were reduced by randomly shifting each ICA component activation time course by a time (*n*–1) x *T*, where *T* is larger than any autocorrelation time (practically, T must be a least one trial length). This random shifting was achieved for each surrogate trial by randomly sampling without replacement from the remaining trials for each ICA activation time course. This yielded effective values for *T* ranging from one to several hundred trial lengths, depending on the number of trials for a given data set. The shifted time courses were then further randomly shifted within a trial in a cyclical manner, such that for *t* time points, the *t+nth* time point is identical to the *nth* time point. This additional step was necessary to remove any lingering dependencies in the categorization task EEG data due to the fact that each visual stimulus may have produced a similar evoked brain response at a particular time on any given trial. Cyclically-shifting the data within a trial ensured that the time points of the ICA component activations reflected different stages of the evoked response, thus further reducing the dependencies among the ICA activations. Although the estimated induced categorization task and resting state EEG signals did not contain evoked responses, the surrogate data for these signals were also created using within-trial cyclical-shifting in time in order to maintain comparability across the analyses of the two kinds of EEG responses. Finally, the shifted ICA activations were transformed back into EEG sensor space via the ICA inverse mixing matrix to yield the surrogate EEG signals.

This surrogate-creation procedure was performed on the original preprocessed EEG signals, *X_i_(t)*, for the resting state data and the estimated induced EEG signals, *X_i_(t)_Induced_*, for the induced categorization task responses. For the evoked categorization task responses, a set of bootstrapped estimations of the evoked response was created by random sampling with the replacement of trials to create a single estimated evoked response [60]. The number of trials in each random sample was equal to the original number of trials for a given participant and task condition, and each set of bootstrapped samples included the observed non-bootstrapped estimation of the evoked response as one of the estimations. The surrogate-creation procedure was then performed on the set of bootstrapped estimations of the evoked response.

*I(X)* and *C_I_(X)* were then computed for each set of surrogate data. The mean and two-tailed 95% confidence intervals (*CIs*) of the surrogate integration and complexity were computed from 50 surrogate data sets created for each participant and data condition. Observed mean complexity or integration values outside of these confidence intervals were interpreted as being significantly different from any integration/complexity arising from spuriously-coincident volume-conducted EEG activity. Surrogate data testing was performed via in-house MATLAB scripts, with ICA decompositions implemented using the extended infomax runica algorithm (with data whitening and default stopping weight change = 1e-07) of the EEGLAB toolbox for MATLAB [40].

### 2.9. Dipole Modeling of Categorization Task EEG Data

The goal of the dipole modeling implemented here was to simulate categorization task alpha-range scalp EEG properties (power, synchronization, integration, and complexity) from a collection of neural sources with known integration and complexity in a manner that included the effects of volume conduction. The simulated EEG signals originated from oscillating, fixed-location intracranial dipole sources in a concentric 4-shell spherical head forward volume-conduction model [34,35,36], with scalp electrode locations identical to those used for the 72-channel EEG recordings. The details of the head model may be found in [11]. In brief, two clusters of 20 radially-oriented dipole generators (40 dipoles total) were created at posterior, extrastriate cortical locations, one cluster in each hemisphere (Figure 4). The locations of these neural generators overlapped with, but were not identical to, those of generators typically activated during visual perception and categorization. However, the goal here was not to provide an exact model of these neurocognitive processes. Instead, the goal was simply to emulate general alpha-range visuocortical activation in a manner that would roughly reproduce the global patterns of task-related EEG properties observed at the scalp [19,31,32,43]. This strategy is supported by the fact that any measured scalp electromagnetic distribution may be described by an infinite number of possible source distributions [61].

One hundred 2 s trials (256 HZ sampling rate) were initially created for each simulation via in-house MATLAB scripts. During each simulated epoch, the magnitude of each dipole source moment oscillated over time as characterized by amplitude, frequency, and phase variations chosen to simulate the observed prestimulus and postimulus alpha-range EEG states. Also, coupling among the posterior visual dipole source oscillators was simulated according to the Kuramoto model [62] using a 4^th^-order Runge-Kutta numerical integration method. The main interest was the central 1 s portion of a trial where the variation in dipole properties occurred; the simulations included the preceding and subsequent 0.5 s portions of the epoch to provide data padding to reduce bandpass-filtering window edge artifacts in the central portion of the trial (see below).

Prestimulus visual dipole generators were assigned uniformly random frequencies (9–11 Hz) and starting phases (0–2π radians), with a constant Kuramoto coupling parameter equal to 20 over the central 1 s portion of each epoch and equal to zero outside of this time range. Prestimulus dipole generator amplitude fluctuations were modeled by multiplying each simulated dipole moment time course by a Gaussian window (*σ_t_* = 125 ms) whose temporal center randomly varied uniformly along the time dimension of each epoch (±750 ms around the epoch center with separate variation for each dipole).

Poststimulus visual dipole generators were also assigned uniformly random frequencies (9–11 Hz). However, these dipoles were divided into two groups, one group (75% of dipoles) modeling a deactivation of non-phase-locked alpha-range activity that typically follows visual stimulation [19,32,43] and a second group (25% of dipoles) that modeled a stimulus-locked increase in alpha activity [19,31,32]. The cortical locations of the activated poststimulus dipole were evenly interspersed among the deactivated dipole locations. The deactivated dipoles were assigned uniformly random starting phases (0–2π radians) and had a Kuramoto coupling parameter that linearly increased from −20 to +10 over the central portion of an epoch (the parameter was equal to zero outside this range). In addition, the amplitude of each deactivated dipole decreased and then increased over the central portion of an epoch following an inverted Gaussian envelope (*σ_t_* = 500 ms), with a randomized initialization of the amplitude change (~500 ± 78 ms) and separate variation for each dipole. The activated dipoles were assigned starting phases that uniformly varied over a small interval (0–π/50 radians) and had a Kuramoto coupling parameter equal to 10 over the first 406 ms of the central portion of an epoch (to simulate a rapid stimulus-locked coupling of the activated oscillators), −20 over the next 406 ms (to simulate a slower decoupling of the activated oscillators), and then equaled zero for all other time points. In addition, the activated dipoles coherently changed in amplitude with an initial increase then decrease according to a Gaussian envelope (*σ_t_* = 750 ms; initialized mean peak latency range ~852 ± 20 ms across simulations, then varied within a simulation by initialized mean peak latency ±39 ms). This was followed by a joint increase from the middle to the end of the central portion of an epoch according to a wide Gaussian envelope (*σ_t_* = 2000 ms). The maximum amplitude of all visual dipole sources was 2 μA-cm.

In addition to the simulated posterior visual dipoles, the remainder of the model cortex was filled with 148 equally spaced dipoles to simulate background cortical processes [11]. These background dipoles had constant dipole source moments of 1 μA-cm, randomly varied in frequency (8–13 Hz), and had randomized nonstationary phases achieved via an FFT-based procedure [58].

Scalp potential topographies and time courses were simulated for each dipole generator separately via forward volume-conduction transformation to the scalp. The final simulated EEG scalp record was constructed from the sum of the individual dipole topographies at each time point, following the linearity of the volume-conduction transformation [36]. Simulated scalp signals were then transformed to Laplacian-based current source densities and bandpass-filtered in the alpha-range. In order to reduce filter edge-artifacts, simulated trials were truncated to the central 1 s portion of an epoch after filtering. Finally, like the observed data, simulated scalp signals were downsampled in space and time before computation of integration, complexity, global field power, and synchronization. Sixteen separate simulations were performed to match the number of participants contributing to the observed data; final values were averaged across simulations.

## 3. Results

All empirical data, stimulus materials, and MATLAB data analysis and dipole simulation scripts are available online via the Texas State University Data Repository [63] at https://dataverse.tdl.org/dataverse/entropyeeg.

### 3.1. Observed Induced EEG Activity Integration and Complexity

Mean observed induced EEG activity integration, *I(X)*, and interaction complexity, *C_I_(X)*, values are given in Table 1; ANOVA results are given in Table 2. For categorization task EEG integration, a main effect of Time Interval showed that *I(X)* increased from the pre- to post-stimulus interval across the two categorization tasks. The main effects of Task and the Task x Time Interval interaction were not statistically significant for categorization task induced *I(X)*; see Table 2.

For categorization task induced EEG complexity, a main effect of Time Interval (Table 2) showed that across the two categorization tasks, *C_I_(X)* decreased from the pre- to post-stimulus interval; see Table 1. The main effects of Task and the Task x Time Interval interaction were not statistically significant for categorization task induced *C_I_(X)*; see Table 2.

For the resting state EEG, the main effects of Resting State (Table 2) for induced *I(X)* and *C_I_(X)* showed that I(X) was larger and *C_I_(X)* was smaller for eyes closed versus eyes open conditions; see Table 1.

The main effects of induced EEG integration and complexity comparing the resting states and categorization task conditions (after collapsing across task type) were significant (Table 2), with all post-hoc *ps__corrected_* < 0.006. This showed that *C_I_(X)* monotonically decreased as *I(X)* monotonically-increased across the four data conditions, consistent with the theoretically predicted pattern for the integration-complexity relationship in the high-integration regime (Figure 1).

Mean surrogate *I(X)* and *C_I_(X)* values are given in the right columns of Table 1. All observed mean integration and complexity values lay outside the surrogate 95% two-tailed confidence intervals for all categorization task and resting state conditions (although the statistical significance of this effect was just over the threshold for the resting state eyes open data). It is thus unlikely that the observed induced integration and complexity are attributable to spuriously-coincident volume-conducted interactions. The categorization task surrogate *I(X)* and *C_I_(X)* values were lower than the observed values, consistent with the theoretical prediction [6] that the randomization involved in the surrogate-data procedure should shift the data into a regime of lower structural order (Figure 1). For the resting state data, the surrogate *I(X)* values were smaller and the *C_I_(X)* values were larger than the observed values, a pattern that has been observed before for resting state data [11]. This pattern can also be explained in terms of a shift of the surrogate data towards lower structural order, but with the observed resting state EEG initially in a higher ordered and integrated state relative to the categorization task EEG that was then shifted by the surrogate procedure into an intermediate regime characterized by higher complexity (see Figure 1).

### 3.2. Observed Evoked EEG Activity Integration and Complexity

Mean observed evoked EEG *I(X)* and *C_I_(X)* values for the categorization tasks are given in Table 3; ANOVA results are given in Table 4. For categorization task evoked EEG integration, a main effect of Time Interval showed that across the two categorization tasks, *I(X)* increased from the pre- to post-stimulus interval. The main effect of Task was not statistically significant, *F*(1,15) = 2.91, *p* < 0.109, η^2^_*P*_ = 0.16, but the Task x Time Interval interaction was significant for categorization task evoked *I(X)*; see Table 4. Decomposition of this interaction showed that evoked *I(X)* did not differ between the two tasks in the prestimulus interval, *post-hoc p__corrected_* < 0.483, but was larger for the 2-Exemplar versus 1-Exemplar task in the poststimulus interval, *p__corrected_* < 0.036; see Table 3.

For categorization task evoked EEG complexity, a main effect of Time Interval (Table 4) showed that across the two categorization tasks, *C_I_(X)* decreased from the pre- to post-stimulus interval. The main effects of Task and the Task x Time Interval interaction were not statistically significant for categorization task evoked *C_I_(X)*; see Table 4.

Mean surrogate *I(X)* and *C_I_(X)* values are given in the right columns of Table 3. All mean observed values lay outside the surrogate 95% two-tailed confidence intervals for all categorization task and resting state conditions (although the statistical significance of this effect was just over the threshold for the 2-Exemplar task poststimulus data). It is thus unlikely that the observed evoked *I(X)* and *C_I_(X)* are attributable to spuriously-coincident volume-conducted interactions. Surrogate evoked *I(X)* and *C_I_(X)* were lower than the observed values, again consistent with the theoretical prediction [6] that the randomization involved in the surrogate-data procedure should shift the data into a regime of lower structural order; see Figure 1.

Finally, the ANOVA comparing induced and evoked *I(X)* after collapsing across the tasks produced significant main effects of EEG Type and Time Interval, and a significant EEG Type x Time Interval interaction (Table 4). Also, the ANOVA comparing induced and evoked *C_I_(X)* after collapsing across the tasks produced significant main effects of EEG Type and Time Interval, and a significant EEG Power Type x Time Interval interaction (Table 4). Decomposition of the interactions showed that whereas there was no difference between induced and evoked EEG properties in the prestimulus interval, *ps_corrected_* < 1, in the poststimulus interval, evoked *I(X)* was greater and evoked *C_I_(X)* was smaller than for the corresponding induced EEG properties, *ps_corrected_* < 0.002.

### 3.3. Observed EEG Power and Synchronization

The time course of EEG global field power and synchronization for both categorization tasks are shown in Figure 5 and Figure 6. Both time courses were dynamic and complex, yet followed a similar morphology between the two categorization tasks. Figure 7 shows the scalp topographies of the alpha-range resting state GFP. The topographies exhibit the standard posterior maximum power observed for eyes closed and eyes open resting states [11,64,65]. Table 5 lists time-averaged induced EEG global field power, *GFP_Induced_*, and synchronization, *Λ_Induced_*, and evoked EEG global field power, *GFP_Evoked_*, and synchronization, *Λ_Evoked_*, for the categorization tasks and for the resting states. ANOVA results are given in Table 6 and Table 7; GEE-based regression results are given in Table 8. 

For categorization task induced EEG power, a Time Interval main effect (Table 6) showed that across the two categorization tasks, *GFP_Induced_* decreased from the pre- to post-stimulus interval; see Table 5. The main effect of Task was not statistically significant for categorization task *GFP_Induced_*, but the Task x Time Interval interaction was significant. This interaction was due to a larger pre-to-post-stimulus induced power decrease for the 2-Exemplar versus 1-Exemplar task, *p* < 0.018; see Table 5.

For categorization task evoked EEG power, a main effect of Time Interval (Table 7) showed that across the two categorization tasks, *GFP_Evoked_* increased from the pre- to post-stimulus interval; see Table 5. A significant main effect of Task (Table 7) showed that *GFP_Evoked_* was larger during the 2-Exemplar Task than the 1-Exemplar Task; see Table 5. The Task x Time Interval interaction was not significant; see Table 7.

The ANOVA comparing the induced and evoked EEG power after collapsing across the tasks (Table 7) produced significant main effects of EEG Component and Time Interval, and a significant EEG Component x Time Interval interaction. These outcomes indicated that *GFP_Induced_* was greater than *GFP_Evoked_* overall (see Table 5), with a stimulus-related decrease in *GFP_Induced_* and increase in *GFP_Evoked_* in the poststimulus interval (as earlier in this section).

The task and time interval-collapsed GEE-based regression analyses (Table 8) relating categorization task EEG induced power to induced *I(X)* and *C_I_(X)* showed that *GFP_Induced_* was positively related to induced *I(X)* and negatively related to *C_I_(X)*. Similarly, *GFP_Evoked_* was also positively related to evoked *I(X)* and negatively related to *C_I_(X)*. Thus, high levels of EEG power corresponded to high levels of EEG integration and low levels of EEG complexity, and vice versa. However, the GEE-based regressions (Table 8) comparing stimulus-related changes (poststimulus–prestimulus differences) in *GFP_Induced_* and *GFP_Evoked_* to changes in *I(X)* and *C_I_(X)* were nonsignificant.

For categorization task EEG synchronization, a main effect of Time Interval (Table 6 and Table 7) showed that across the two categorization tasks, *Λ_Induced_* decreased and *Λ_Evoked_* increased from the pre- to post-stimulus interval; see Table 5. The Task main effect was not significant for both synchronization measures. The Task x Time Interval interaction was not significant for *Λ_Evoked_*, but was significant for *Λ_Induced_*. Although decomposition of the latter interaction suggested that *Λ_Induced_* for the 2-Exemplar task was larger than for the 1-Exemplar task in the pre-stimulus interval (Table 5), this difference was only marginally significant, *p_corrected_* < 0.06.

The ANOVA comparing the induced and evoked EEG synchronization after collapsing across the tasks (see Table 7) produced significant main effects of EEG Type and a significant EEG Component x Time Interval interaction, but the Time Interval main effect was nonsignificant; see Table (AAA). These outcomes indicated that *Λ_Evoked_* was greater than *Λ_Induced_* overall, with a stimulus-related decrease in *Λ_Induced_* and increase in *Λ_Evoked_* in the poststimulus interval (as reported earlier in this section); see Table 5.

The task and time interval-collapsed GEE-based regression analyses (Table 8) relating categorization task induced EEG synchronization to induced *I(X)* and *C_I_(X)* showed that *Λ_Induced_* was negatively related to induced *I(X)* and positively related to induced *C_I_(X)*. The relationships between *Λ_Evoked_* and *I(X)* and *C_I_(X)* were nonsignificant. Thus, high levels of induced EEG synchronization corresponded to low levels of EEG integration and high levels of EEG complexity, and vice versa. The GEE-based regressions comparing stimulus-related changes in *Λ_Induced_* and *Λ_Evoked_* to changes in *I(X)* and *C_I_(X)* were nonsignificant; see Table 8.

Finally, for the resting state EEG, main effects of Resting State for *GFP_Induced_* and *Λ_Induced_* (Table 6) showed that both measures were larger for eyes closed versus eyes open conditions; see Table 5. The main effect of induced EEG global field power comparing the resting states and categorization task conditions was also significant (Table 6). In general, *GFP_Induced_* was smaller for the categorization task conditions versus the resting states, *ps_corrected_* < 0.006; see Table 5. The main effect comparing *Λ_Induced_* across the resting states and categorization task conditions was also significant (Table 6). Post-hoc testing showed a significant *Λ_Induced_* difference between post-stimulus versus eyes closed conditions, *p_corrected_* < 0.032, and between pre-stimulus versus eyes open conditions, *p_corrected_* < 0.045, (see Table 3), but no other significant differences between the categorization task and resting states, *p_corrected_* < 0.434.

### 3.4. Categorization Task Behavior

Participants were more accurate at stimulus categorization in the 1-Exemplar task (median task accuracy: 97%) versus the 2-Exemplar task (median task accuracy: 80%), Wilcoxon Signed-Ranks test *W* = 123, *z* = 2.85, *p* < 0.005, *r* = 0.71. Also, participants indicated their stimulus categorizations more rapidly in the 1- Exemplar task (mean reaction time: 398 ms ± 13 ms) versus the 2- Exemplar task (mean reaction time: 492 ms ± 23 ms), *F*(1,15) = 14.30, *p* < 0.002, η^2^_*P*_ = 0.49.

The GEE-based regressions comparing participant accuracy to stimulus-related changes (poststimulus–prestimulus differences) in induced *I(X)* were not significant, *ps_corrected_* > 0.28. However, the regression comparing participant task accuracy to stimulus-related changes in induced *C_I_(X)* showed a negative relationship between stimulus-related changes in induced *C_I_(X)* and task accuracy, *β* = −0.21 ± 0.10, *Wald* χ^2^ (1,16) = 4.20, *p* < 0.004. Also, the regression comparing participant task accuracy to stimulus-related changes in evoked *I(X)* showed a negative relationship between stimulus-related changes in evoked *I(X)* and task accuracy, *β* = −0.47 ± 0.15, *Wald* χ^2^ (1,16) = 10.05, *p* < 0.002. There was no significant relationship between accuracy and evoked *C_I_(X)*, *ps* < 0.105.

The GEE-based regressions comparing participant reaction time to stimulus-related changes in induced *I(X)* and *C_I_(X)* were not significant, *ps_corrected_* = 1. However, there was a significant Task X Evoked EEG integration interaction for the relationship between evoked *I(X)* and task reaction time, *Wald* χ^2^ (1,16) = 10.48, *p* < 0.001. Decomposition of this interaction indicated a significant positive relationship between reaction time and evoked *I(X)* during the 2-Exemplar task, *β* = 0.64 ± 0.12, *Wald* χ^2^ (1,16) = 29.74, *p_corrected_* < 0.002. A Task x Evoked *C_I_(X)* interaction was also significant for reaction time, *Wald* χ^2^ (1,16) = 3.86, *p* < 0.049, but the follow-up tests were nonsignificant, *ps_corrected_ < 0.144*.

### 3.5. Dipole Simulation of Categorization Task EEG

The development of the dipole simulations of the empirically-observed categorization task data involved the initial simulation of basic EEG signals that varied in terms of the dipole source amplitude, phase, and synchronization of dipole oscillations and common changes in the amplitude; these initial simulations are described in the Appendix A. The purpose of these initial simulations were to understand how EEG properties of interest varied with simple changes in the dipole amplitude and synchronization. The understanding gained from these initial simulations enabled the creation of more realistic simulations of categorization task EEG properties.

Figure 8 shows the time courses of EEG global field power and synchronization averaged across 16 separate simulations of the categorization task EEG data. For the most part, the simulated time courses followed a dynamic and complex morphology similar to that observed for the empirical categorization task data. One exception to this similarity is the simulated evoked EEG responses. Here, in the early part of the time epoch, evoked global field power displayed a greater degree of oscillation and evoked synchronization showed a greater magnitude relative to the empirical data. Nevertheless, the general pattern of pre- versus poststimulus differences in all EEG properties was similar to that of the observed categorization tasks and was highly reliable across the 16 simulations. This is evidenced in Table 9, which shows time-averaged information-theoretic, spectral power, and synchronization EEG properties for the simulated categorization task data, and ANOVA results conducted across the 16 separate simulations. This supports the conclusion that the empirically-observed *I(X)* and *C_I_(X)* patterns are not mere artifacts of volume conduction, but reflect true dynamical changes in integration and segregation among neuronal networks in the brain.

### 3.6. Comparison of KNN-based and Gaussian-Based EEG Integration and Complexity Estimation

An important issue of interest is how the KNN-based entropy estimation of *I(X)* and *C_I_(X)* compares to estimation computed using Gaussian-based entropy as has been performed in previous studies [9,11,16,17]. This issue was examined here by first transforming the dimension-reduced categorization task, resting state, and dipole simulation EEG data to follow a Gaussian distribution [11,66]. Distributional testing [67,68,69] verified that the dimension-reduced EEG data significantly deviated from normality for all subjects before the Gaussian-transformation and conformed to normality afterwards; see Appendix A. Then, *I(X)* and *C_I_(X)* were each computed from the Gaussian-transformed data using the KNN-based estimators and Gaussian-based estimators [6,17,70,71,72,73,74] (see Appendix A for a mathematical description of the Gaussian entropy estimator). The KNN estimation of the Gaussian-transformed data served as an intermediate step that allowed a better examination of the effect the Gaussian-transformation had on the statistical characteristics of the data.

The quantitative and statistical results of this analysis are reported in the Appendix A. All observed mean *I(X)* and *C_I_(X)* values lay outside the surrogate 95% two-tailed confidence intervals for all categorization task and resting state conditions. The differences between observed and surrogate values were similar to those found for the non-Gaussian-transformed EEG data (see Section 3.1 and Section 3.2, above), again following theoretical predictions and previous observations [6,11]; see Figure 1. It is thus unlikely that the integration and complexity values computed from the observed Gaussian-transformed data are attributable to spuriously-coincident volume-conducted interactions.

Nevertheless, in general, different patterns of between-condition *I(X)* and *C_I_(X)* differences were found for the Gaussian-transformed data than for the non-transformed data, regardless of the kind of entropy estimation used to compute *I(X)* and *C_I_(X)*. In the case of the observed categorization task EEG data, both forms of entropy estimation yielded smaller *I(X)* values and larger *C_I_(X)* values for post stimulus versus prestimulus induced Gaussian-transformed EEG data (although the complexity differences were not statistically significant). This same pattern of EEG integration and complexity differences was also observed for the Gaussian-transformed EEG simulations computed using the Gaussian entropy estimator. The directions of all of these differences were opposite to that observed for the corresponding non-transformed data. The direction of the KNN-based *C_I_(X)* differences computed from the induced Gaussian-transformed simulated data was also opposite to that computed for the non-transformed data.

For the evoked Gaussian-transformed empirical and simulated EEG data, both forms of entropy estimation yielded larger *I(X)* and *C_I_(X)* values for post stimulus versus prestimulus intervals. The directions of the EEG complexity differences were opposite to that observed for the corresponding non-transformed data.

For the resting state data, the KNN-based estimation produced larger *I(X)* and *C_I_(X)* values for eyes closed versus eyes open Gaussian-transformed EEG data. The direction of the EEG complexity differences were opposite to those observed for the corresponding non-transformed data. However, the use of the Gaussian-based estimator yielded the same overall pattern as for the non-transformed data using the KNN-based estimator (i.e., larger *I(X)* and smaller *C_I_(X)* values for poststimulus versus prestimulus intervals), a pattern that is also consistent with previous observations [11].

## 4. Discussion

### 4.1. Stimulus-Induced and Stimulus-Evoked Changes in Scalp EEG Integration and Complexity

The main goal of the present study was to utilize KNN-based entropy estimation to quantify stimulus-elicited EEG integration *I(X)* and interaction complexity *C_I_(X)* of human alpha-range EEG signals. The EEG signals were acquired during two simple visual categorization tasks that varied in terms of the complexity of the category structure and the number of stimulus exemplars per category. It was found that the KNN-based measures reliably discriminated between pre-stimulus ongoing EEG states and post-stimulus-induced and evoked EEG states in a manner consistent with the theoretical relationship between integration and complexity [6,11]. Dipole modeling and surrogate statistical analysis showed that scalp-level EEG integration and complexity more likely reflected the functional segregation and integration of activity of the brain rather than spurious-correlations due to volume conduction. This conclusion is bolstered by the use of a current source density transformation of the EEG data (see Section 2.4, above), which reduces the effects of volume conduction in EEG integration and complexity measures [11].

Across both categorization tasks, *I(X)* increased and *C_I_(X)* decreased with stimulus-onset relative to the prestimulus interval for both induced and evoked EEG responses. However, these stimulus-related changes were greater for evoked versus induced EEG responses. One possibility for this difference is that the induced EEG response may not only contain non-stimulus-locked EEG signals that are reliably elicited in response to stimulus events, but also unreliable background brain activity not related to stimulation per se (such background activity is averaged out during the computation of the evoked EEG signal; see Equations (8) and (9). It may be that the random nature of the background activity biased the poststimulus induced response toward a lower integration/higher complexity regime (Figure 1). Alternatively, it may be that the brain networks underlying the evoked EEG response are more global in nature than for the induced EEG response and thus unify a larger number of local networks distributed across the brain. This would lead to a larger increase in the evoked poststimulus integration and a decrease in the evoked poststimulus complexity. A third possibility is that both the large background activity and a greater integration of the evoked response are responsible for the present *I(X)* observations. Testing these competing hypotheses is a subject for future research.

Evoked *I(X)* also distinguished between the easier 1-Exemplar categorization task and the more difficult 2-Exemplar task. Evoked *I(X)* was greater for the 2-Exemplar task versus the 1-Exemplar task and had a negative relationship to the categorization accuracy across both tasks. Evoked *I(X)* was also negatively related to reaction time in the 2-Exemplar task, with larger EEG integration associated with longer stimulus categorization reaction times. These latter findings are evidence for a connection between cortical integration and categorization task behavior, with behavioral performance negatively affected by increases in the integration of evoked brain responses. Future research should address how *I(X)* and *C_I_(X)* may vary with respect to behavioral performance during more difficult and complex categorization tasks than those utilized here.

Analysis of the EEG spectral power and synchronization showed that absolute levels of induced and evoked EEG integration and complexity were related to absolute levels of induced and evoked EEG power and induced EEG synchronization. In general, high levels of EEG power corresponded to high levels of EEG integration and low levels of EEG complexity, and vice versa. Similarly, high levels of induced EEG synchronization corresponded to low levels of induced EEG integration and high levels of EEG complexity. However, stimulus-related changes in induced and evoked *I(X)* and *C_I_(X)* were not related to stimulus-related changes in the corresponding EEG power and synchronization. This suggests that although stimulus-related changes in these EEG properties occur simultaneously, *I(X)* and *C_I_(X)* changes reflect different functional processes than EEG power and synchronization changes, in particular the coordinated interaction among neural hierarchies necessary for stimulus processing. Nevertheless, the present findings of stimulus-related changes in *I(X)* and *C_I_(X)* computed from alpha-range EEG signals are consistent with previous findings of a relationship of this EEG frequency band to visual perception [19,31,32].

To the author’s knowledge, the present study is the first time EEG integration and complexity have been analyzed separately for induced and evoked EEG signals and for brain responses related to visual categorization. The present study is also the first time these information-theoretic brain measures have been analyzed using KNN entropy estimation. More generally, these findings provide additional evidence for a role of brain integration and complexity in visual perception [12], specifically the integration and complexity of brain networks mediating changes in visually-elicited induced and evoked alpha-range oscillations. In the present study, the EEG power and synchronization analyses applied to the observed and simulated data indicated that induced alpha power and synchronization decreased with visual stimulation, whereas evoked alpha power and synchronization increased (see Section 3.2 and Section 3.4). These findings are consistent with previous observations for visually-elicited EEG responses. Visually-elicited changes in induced alpha-range EEG activity are believed to reflect increases in cortical excitation [20,32,43,75,76], whereas such changes in evoked alpha-range EEG activity are believed to reflect inhibition in cortical activity [31,32,43,75]; the latter may serve to increase the signal-to-noise ratio for visuocognitive processes occurring later in time [32]. Moreover, prestimulus alpha-range oscillations also reflect ongoing attentional vigilance to visual input [76]. The present study advances our understanding of the functional relevance of alpha-range EEG to visual perception by showing that the brain networks undergoing visually-elicited cortical excitation and inhibition also become more integrated and less complex, a phenomenon that has implications for our understanding of brain dynamics more generally (see Section 4.2 below).

Moreover, to the extent the visual feedback of the present categorization tasks was sufficiently motivational (see Section 2.3, above), the present findings may be relevant for understanding the role of brain integration and complexity in the cognitive processing of visual stimuli leading to penalty and reward [18]. However, it should be noted that the present study utilized an entropy metric (the K-NN entropy estimator) that is superior for stimulus-elicited brain responses than the Gaussian-based metric used in previous studies and thus likely provides more accurate stimulus-elicited patterns of EEG integration and complexity (for further discussion on this issue, see Section 4.3, below).

### 4.2. Resting State Scalp EEG Integration and Complexity

In the present study, induced *I(X)* was greater and *C_I_(X)* was smaller for eyes closed versus eyes open resting state conditions. The EEG integration finding replicates the observations of several previous studies that used Gaussian-based entropy estimators to compute *I(X)* and *C_I_(X)* [9,11,16,17]. The present EEG complexity finding also replicates the observation of one of these previous studies [11], but shows an opposite pattern of eyes closed versus eyes open EEG complexity relative to the other studies.

However, as pointed out by Trujillo and colleagues in [11], it is likely that this discrepancy is due to the fact that the Gaussian-based estimator for *H* is highly dependent on the degree to which the data meet Gaussian statistical assumptions. The prior studies assumed, but did not verify, that their resting state EEG data was univariate and multivariate Gaussian-distributed, an assumption that is often partially or completely violated for physiological signals [23,77,78,79]. In contrast, Trujillo et al. found their EEG signals to substantially deviate from normality and applied an algorithm [66] to transform the non-normal EEG distribution to Gaussian, the success of which was then verified through appropriate distributional testing. Trujillo et al. found that the computation of resting state *C_I_(X)* depended on whether or not the data was Gaussian-distributed. Prior to the transformation of the data, Trujillo et al. found smaller EEG complexity during the eyes open vs. eyes closed resting state conditions, consistent with the previous studies making the Gaussian assumption [9,16,17]. After the Gaussian transformation, Trujillo et al. observed a larger complexity for eyes open versus eyes closed conditions, as observed here. Trujillo et al. did not find the computation of resting state *I(X)*, or the computation of the power and synchronization of resting state EEG signals, to be greatly affected by the Gaussian transformation.

Given the findings of [11], and the fact that the KNN-estimators used here are nonparametric and thus free of distributional assumptions, it is likely that the present estimations of smaller *C_I_(X)* for eyes closed versus eyes open resting state conditions are more accurate with respect to the true state of the human brain during these ongoing wakeful states. Moreover, the presently observed EEG complexity pattern is consistent with the theoretical expectation of a decrease in EEG complexity in the high integration regime (Figure 1), a regime normally occupied by ongoing scalp-level EEG as shown by [11] via similar dipole modeling techniques as utilized here. Thus, the convergence of the present resting state *I(X)* and *C_I_(X)* findings across two different entropy estimation methods validates the application of KNN-based entropy estimation for quantifying the statistical dependence among ongoing multivariate EEG signals. This convergence also supports the conclusion that eyes closed resting states are characterized by high integration and low complexity relative to eyes open resting states.

One key observation of the present study is that induced EEG complexity monotonically-decreased and induced EEG integration monotonically-increased across the four data conditions under investigation, starting from categorization task prestimulus to poststimulus time intervals and then to eyes open to eyes closed resting states. This behavior is consistent with the theoretically predicted pattern for the integration-complexity relationship in the high-integration regime (Figure 1). The observed range of complexity for these brain states is also consistent with theoretical arguments and experimental evidence that brain dynamics inhabit an extended region of criticality–an optimal balance between ordered and disordered states [80,81,82,83,84]. Criticality is thought to provide the necessary conditions for the brain to implement flexible and efficient information processing while operating in a relatively stable state [80,82,83,85]. Although criticality is believed to be a hallmark of brain resting states [86,87,88,89,90,91,92], in the present study, *C_I_(X)* was highest during the interval of ongoing EEG immediately prior to stimulus presentation during the categorization task. One possible explanation for this finding is that criticality enables the brain to maintain a high sensitivity to external stimulation [80,83,84], a sensitivity that would be maximal during the time in between stimulus presentations when participants had to maintain a constant vigilance for the onset of visual stimuli and constant preparedness to assess the stimulus categories. Criticality is typically indexed via the presence of *1/f* power laws in the distribution of observable brain activity [80,81,93]. However, the present findings suggest that *C_I_(X)* may provide an alternate means to assess the criticality of the brain, especially given methodological difficulties in accurately estimating power laws from empirical data [93]. A study of the relationship between EEG complexity and EEG power law measures would be a fruitful study for future research into the criticality of brain dynamics.

### 4.3. Advantages and Disadvantages of KNN Metrics for EEG Entropy Estimation

As discussed throughout this article, previous studies have utilized Gaussian-based entropy estimators to compute EEG integration and complexity. The advantages of the Gaussian-based entropy estimator are that it is robust, less prone to sampling bias, and computationally-efficient [72,94]. However, the KNN entropy estimators used in the present study have two major advantages over Gaussian-based estimates. First, the KNN estimators are robust in the presence of variance-nonstationary data [24], such as EEG signals over short time periods [22]. Nevertheless, one must take steps to ensure that the data to which one applies the KNN estimators are variance-nonstationary. In the present study, this was achieved by filtering the EEG data to remove trends, spikes, and other data characteristics that reflected nonstationarities (e.g., mean- and trend-nonstationatrity) other than variance-nonstationarity (see Section 2.4, above).

Second, the present KNN entropy estimators are nonparametric in the sense that they do not require the data to follow a specific (normal) statistical distribution and thus avoid the need to perform a Gaussian transformation on the EEG data. The present study found that this transformation changed the statistical, and thus entropic, properties of the EEG data (see Section 3.5, above). The Gaussian-transformed categorization task EEG data exhibited a different pattern of *I(X)* and *C_I_(X)* than the original non-transformed data. This was the case regardless of whether entropy was estimated using the KNN-based or Gaussian-based approach. In contrast, Gaussian-transformation of the resting state EEG data in conjunction with the use of a Gaussian entropy estimator yielded the same basic pattern of *I(X)* and *C_I_(X)* as observed when applying the KNN-based estimator to the original non-transformed resting state data. These findings suggest that, in certain cases, changing the distribution of the EEG data away from its true distribution significantly distorts the entropy estimation away from the true entropy values. This may have occurred for the categorization task EEG data because such signals are inherently nonstationary. Thus, one may conclude that KNN-based estimation should be utilized for computing EEG integration and complexity when the data distribution is unknown or non-normal, and/or when the data is nonstationary. The only time such data transformations should be used is when it can be shown that it does not make a difference, as was the case for the resting state data when using the Gaussian-based entropy estimator (see also Section 4.2, above). (One drawback of the Gaussian transformation used here is that it is a univariate transformation performed on an electrode-by-electrode basis. A multivariate Gaussian distribution was achieved via this procedure, but any preexisting interelectrode dependencies were likely weakened and thus the estimates of *I(X)* and *C_I_(X)* were likely distorted to some degree. Although it is possible to use Cholesky decomposition to implement a Gaussian transformation of multivariate data that preserves data covariance, this technique may fail when the covariance matrix is nearly singular due to high interdependencies among variables [95], such as is often observed for EEG signals.)

A major disadvantage of using KNN entropy estimators to estimate EEG integration and complexity is that even though they are relatively efficient to compute for small multivariate data spaces [26,29], they are computationally intensive to compute for the large data spaces typical of EEG recordings. This is mainly due to the need to perform a SVD for each time point of an EEG epoch as part of the G-KNN computation (Equation (6)) of the joint entropy, *H(X)*, and conditional entropy, *H(X_i_|X–X_i_),* and the need to compute the conditional entropy multiple times during the computation of *C_I_(X)* (see Equation (2)). In a pilot exploration of the KNN-based analysis presented here, it was possible to compute *I(X)* and *C_I_(X)* for the original 72-channel by 256 sample epochs in a reasonable time frame; for the non-surrogate data reported in this paper, this took approximately 4 days using parallel algorithms implemented on a 16-core 2.2 gHz, 64 GB RAM Dell workstation. However, the estimated time required to compute the surrogate data for the full data set was prohibitive (approximately 200 days of computation time). Hence, the need for the spatial and temporal downsampling procedure utilized in the present study *I(X)* and *C_I_(X)* (see Section 2.4, above), which reduced the necessary computation time by a factor of ~25.

Fortunately, initial exploration of the downsampling procedure did not alter the pattern of *I(X)* and *C_I_(X)* observed without the downsampling; hence, the results from the downsampled data were reported in this paper to maintain consistency across surrogate and non-surrogate analyses. Nevertheless, for researchers with access to supercomputing facilities, the computationally-intensive nature of KNN entropy estimation for large data spaces may be negligible in the face of the more accurate EEG integration and complexity estimations that are more likely to be produced when using a full data space. However, for researchers without access to supercomputing facilities, downsampling is required to obtain results in a realistic timeframe, although Gaussian-based estimators may still be used in certain cases (e.g., ongoing resting state data transformed to be approximately Gaussian per [11]) with reasonably accurate results.

### 4.4. Study Limitations

One limitation of the present study is that the reported EEG integration and complexity findings represent aggregate measures pooled across time for each EEG epoch. This is because *I(X)* and *C_I_(X)* were computed by treating each set of EEG signals as a jointly-distributed space of variables, with each time point as an individual observation drawn from the joint-distribution (see Section 2.6), and EEG entropy estimation is based on the probabilities of multielectrode configurations across time within a particular trial. This time-pooled approach precluded a more refined analysis of temporal changes in *I(X)* and *C_I_(X)* that occurred as the stimulus-elicited brain response unfolded over time. One way to achieve a more temporally-refined analysis would be to divide each EEG epoch into small sequential windows across time, as was done for the analysis of EEG complexity by [18] (although in that study, the windows were eventually averaged together to yield an aggregate measure). Nevertheless, there are two drawbacks to this approach. First, the sizes of such sequential windows are necessarily limited to be no smaller than the size of the *kth* nearest neighborhood, which must at least equal the dimension of the joint variable space of the data under analysis for the G-KNN entropy estimator [29] (see Section 1.3 and Section 2.6, above). Thus, the resolution of such a temporal analysis is limited by the number of EEG electrodes composing the data space. Second, this procedure will necessarily increase the overall computation time of *I(X)* and *C_I_(X)* by a factor equal to the number of time windows.

An alternative approach to achieving a temporally-refined analysis would be to treat each set of EEG signals as a jointly-distributed space of variables across trials at each time point. Here, a trial would be treated as an individual observation drawn from the joint-distribution and EEG entropy estimation would be based on the probabilities of multielectrode configurations across trials at a particular time. However, it is unclear if this trial-pooled approach would index the same variability of the data as pooling data across time. When pooling across time, *I(X)* and *C_I_(X)* reflect the degree to which sensors are dependent upon each other at any time point within a trial. In contrast, when pooling across trials, *I(X)* and *C_I_(X)* reflect the degree to which sensors are dependent upon each other at a specific time point across trials. Determining the best method to compute the time-course of stimulus-elicited EEG integration and complexity is an important topic for future research.

A second limitation of the present study is that only a single value of the nearest neighbor parameter, *k*, was utilized to compute the KNN entropy estimates; no attempt was made to investigate the effects of different *k* parameter values on the entropy estimates. As discussed in more detail in Section 2.6, above, the specific value of the *k* parameter utilized here (*k* = 36) was chosen because the value of this parameter should be small to provide a local estimate of the geometry of the joint variable data space, but it should also at least be equal to the number of dimensions of the data space to correctly compute the local ellipsoidal neighborhood around each data point by SVD for the G-KNN entropy estimate [29]. Given that the data space analyzed via the KNN measure had 36 dimensions, the choice of *k* = 36 met both of these criteria (i.e., it was the smallest value of *k* that was equal to or larger than the dimension of the data space). These two criteria (small *k*, but equal to or larger than the data space dimension) are general criteria that are applicable to and appropriate for any data set. An extensive testing of the effects of larger values of *k* was not conducted because (1) it would violate the first of these two necessary criteria; (2) it was computationally-intractable to do so; and (3) because larger values of *k* tend to produce less accurate entropy estimates (although the variance of such estimates is reduced with larger *k* [41,42]). Nevertheless, it may be that larger *k* values will produce more robust between-condition effects in the context explored here (human brain EEG signals); this should be explored in future research.

A third limitation of the present study is that the dipole simulations conducted to provide benchmark scalp-level *I(X)* and *C_I_(X)* measures were phenomenological in the sense that (1) they did not directly simulate the interactions through which interdependent neural relationships are formed in the brain; and (2) the simulated neural generators were rough approximations to the generators typically activated during visual perception and categorization (see Section 2.9, above). The goal here was to simply emulate general alpha-range visuocortical activation in a manner that would roughly reproduce the global patterns of task-related EEG properties observed at the scalp [19,31,32,43] in order to achieve a basic understanding of the integration and complexity of stimulus-elicited brain responses. Nevertheless, although future studies should utilize better models of the neurocognitive processes under investigation, it is likely that the present findings will generalize to other tasks that engage similar stimulus-elicited EEG activity.

A fourth limitation of the present study is the focus on *I(X)* and *C_I_(X)* computed for alpha-range EEG activity. Alpha-range activity was the focus here because of its functional relevance to visual perception and the fact that alpha activity dominates the intrinsic activity of the cortex [96,97,98]. However, other frequency ranges, such as theta-band (4–7 Hz) and beta band (14–30 Hz), also reliably correlate with visual perception and other cognitive processes [76,99]. Future research should examine the EEG integration and complexity of brain networks operating in these other functionally-relevant frequency ranges.

## 5. Conclusions

In conclusion, the present study utilized nonparametric *k-th* nearest neighbor (KNN) differential entropy estimators to characterize the integration and complexity of alpha-range (8–13 Hz) induced and evoked stimulus-elicited brain states during visual categorization. *I(X)* increased and *C_I_(X)* decreased with stimulus-onset relative to the prestimulus interval for both induced and evoked EEG responses, with greater stimulus-related changes for evoked EEG responses. Also, evoked *I(X)* was greater for the difficult versus easier categorization task and was negatively related to categorization performance across both tasks. Although absolute levels of *I(X)* and *C_I_(X)* were related to absolute levels of alpha-range EEG power and synchronization, stimulus-related changes in these measures were largely independent of each other. KNN-based entropy estimation applied to ongoing alpha-range wakeful EEG states replicated previous *I(X)* and *C_I_(X)* findings determined via Gaussian-based entropy estimators. Taken together, these findings support the hypothesis that visual perception and ongoing wakeful mental states emerge from the complex and dynamical interaction of segregated and integrated brain networks operating near an optimal balance between order and disorder.

## Figures and Tables

**Figure 1 entropy-21-00061-f001:**
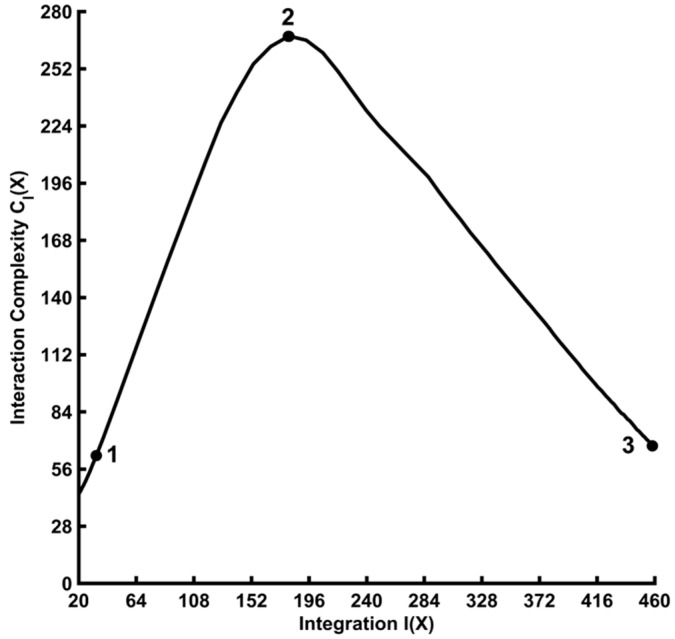
The theoretical relationship between integration, *I(X)*, and interaction complexity, *C_I_(X)*. In this example, *I(X)* and *C_I_(X)* were computed using the KNN-based entropy methods described in this paper applied to simulated 1000 ms epochs of multivariate Gaussian time courses (*n =* 36, *μ =* 0, Toeplitz covariance matrices with increasing intervariable dependencies). The plotted curve reflects average values (in units of bits of information) across 10 separate simulations (100 trials per simulation). Points 1, 2, and 3 on the curve exemplify increasing levels of multivariate statistical dependence; see text of Section 1.1 for further description.

**Figure 2 entropy-21-00061-f002:**
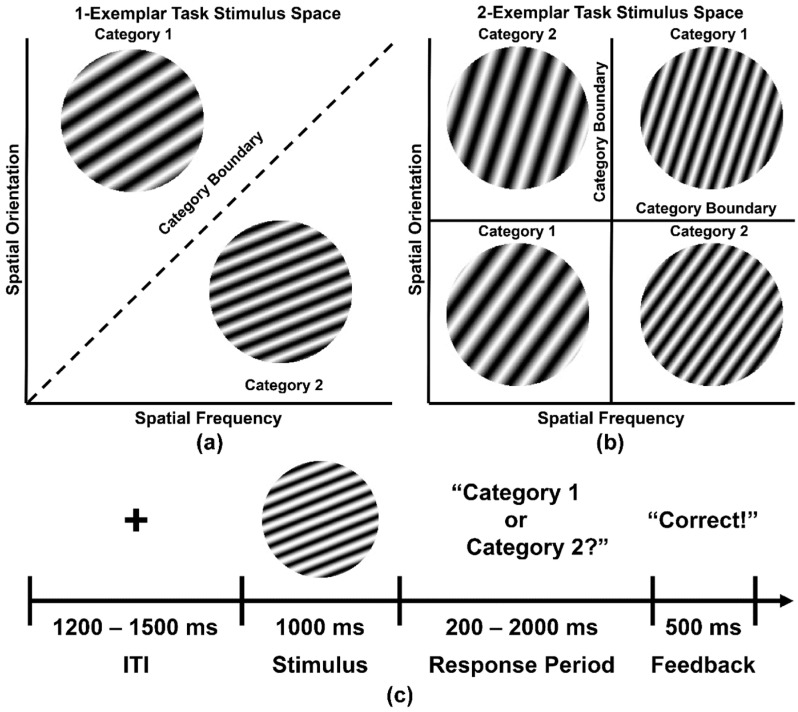
Example category distributions for the (**a**) 1-Exemplar and (**b**) 2-Exemplar categorization tasks; (**c**) Basic categorization task protocol (see Section 2.3 for details).

**Figure 3 entropy-21-00061-f003:**
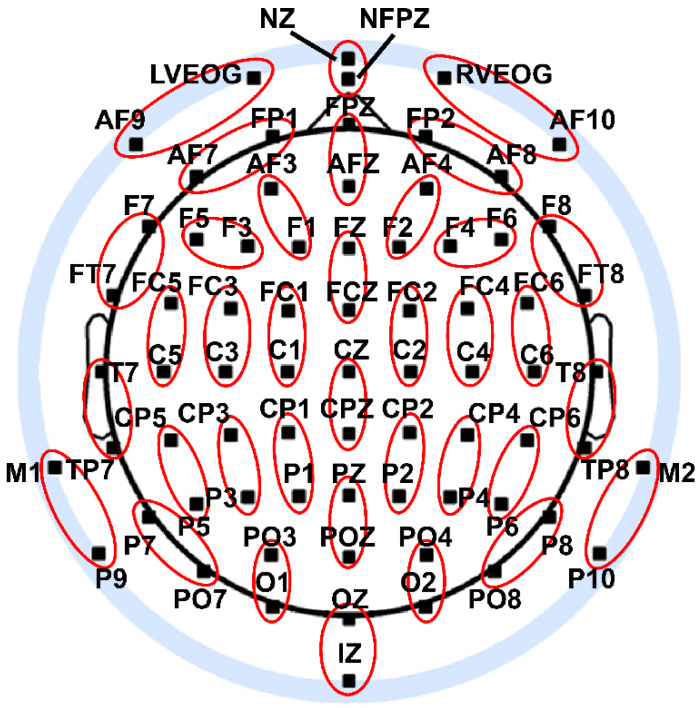
Extended 10–20 scalp locations of EEG recording electrodes. Sites outside the radius of the head represent locations that are below the equator (FPZ-T7-T8-OZ plane) of the (assumed spherical) head model. Light red ovals indicate neighbor electrode pairs across which EEG signals were averaged to implement spatial downsampling during data preprocessing (see Section 2.4).

**Figure 4 entropy-21-00061-f004:**
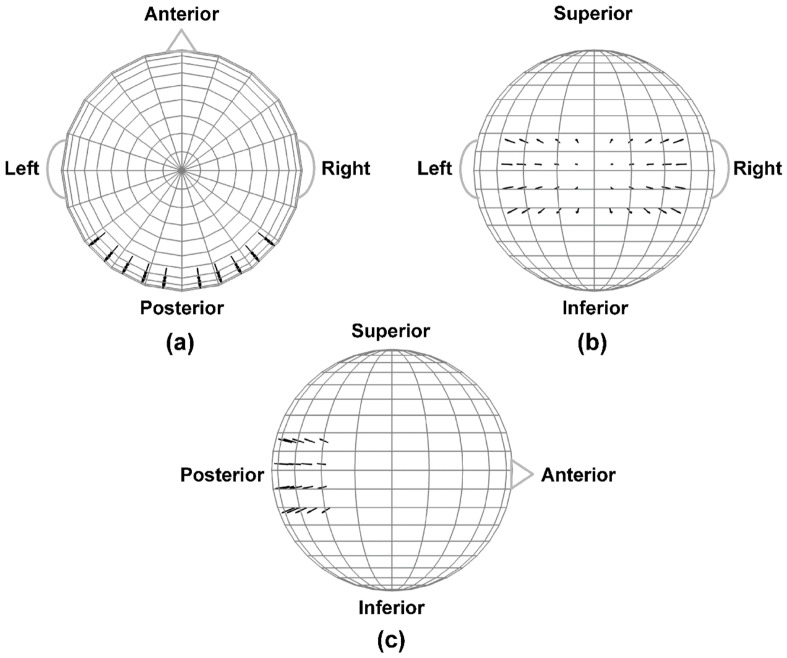
The 4-shell spherical head forward volume-conduction model (only the outer shell surface is represented in the figure). (**a**) Top head view; (**b**) Rear head view; (**c**) Side head view. Black arrows indicate the locations and orientations of the extrastriate dipole sources. The remainder of the spherical surface was filled with 148 equally-spaced background activity dipoles (not shown). All dipoles were placed at superficial (2mm subdural) cortical locations. Figure adapted from [11] with permission of the authors.

**Figure 5 entropy-21-00061-f005:**
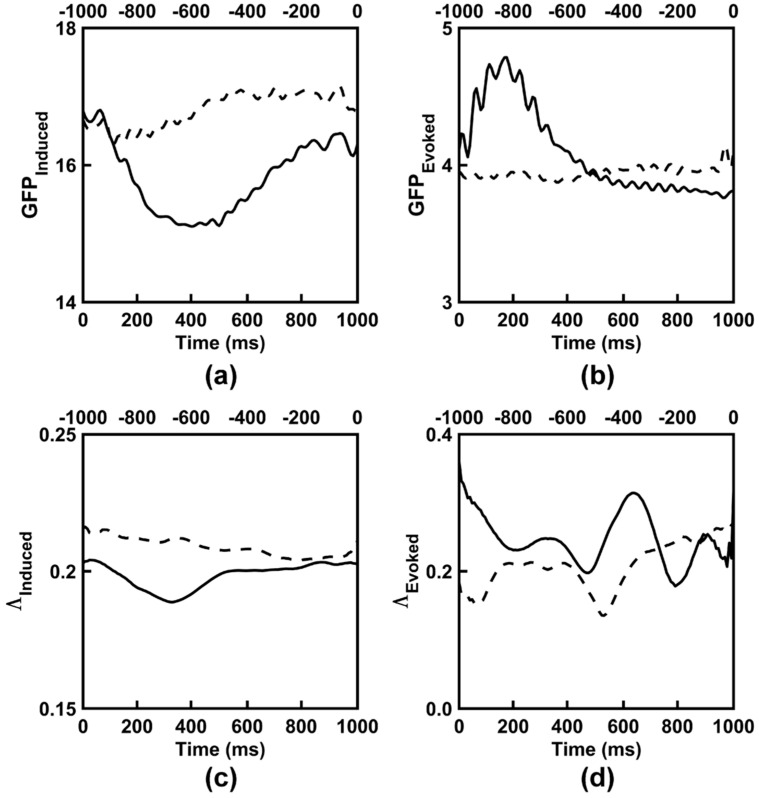
Time courses of EEG power and synchronization for the 1-Exemplar task. (**a**) Induced GFP; (**b**) Evoked GFP; (**c**) Induced synchronization order parameter, *Λ*; (**d**) Evoked synchronization order parameter, *Λ*. Dashed lines = prestimulus responses; solid lines = poststimulus responses. Bottom abscissa scale indicates time points for poststimulus responses relative to stimulus onset; top abscissa scale indicates time points for prestimulus responses. GFP values are in *μV*, order parameters are dimensionless.

**Figure 6 entropy-21-00061-f006:**
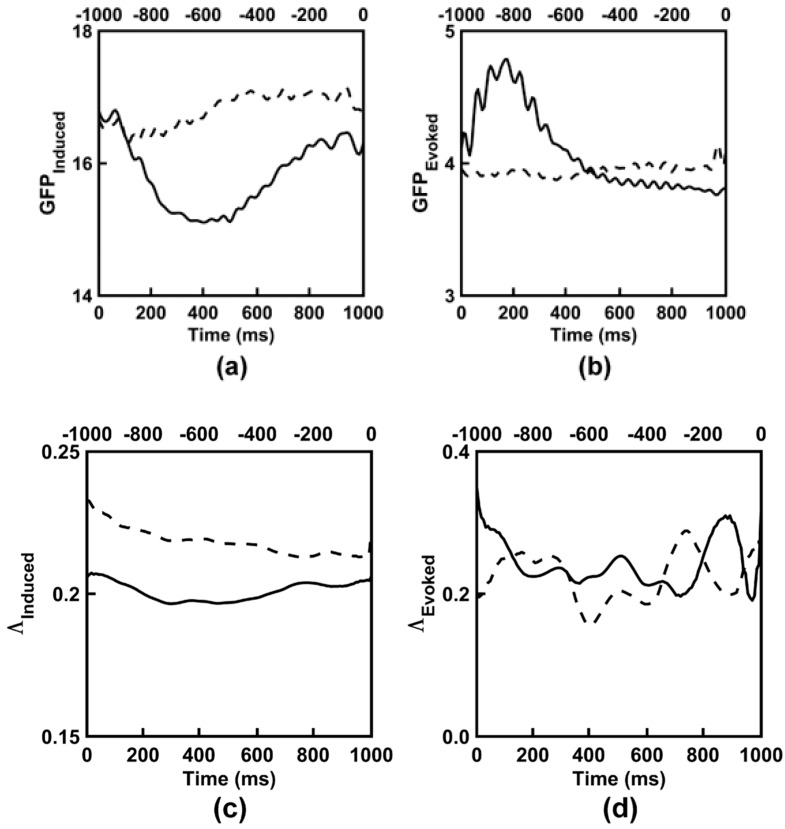
Time courses of EEG power and synchronization for the 2-Exemplar task. (**a**) Induced GFP; (**b**) Evoked GFP; (**c**) Induced synchronization order parameter, *Λ*; (**d**) Evoked synchronization, *Λ*. Dashed lines = prestimulus responses; solid lines = poststimulus responses. Bottom abscissa scale indicates time points for poststimulus responses relative to stimulus onset; top abscissa scale indicates time points for prestimulus responses. GFP values are in *μV*, order parameters are dimensionless.

**Figure 7 entropy-21-00061-f007:**

Head maps showing average alpha-range (8–13 Hz) spectral power scalp topographies for each resting state condition and their between-condition difference. Light/dark color range indicates minimum to maximum values.

**Figure 8 entropy-21-00061-f008:**
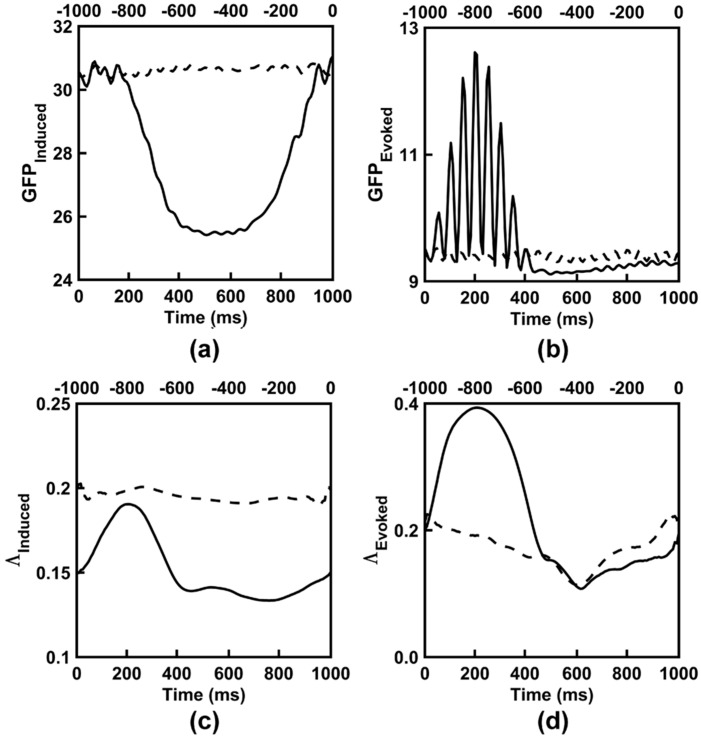
Time courses of EEG power and synchronization for the simulated categorization task data. (**a**) Induced GFP; (**b**) Evoked GFP; (**c**) Induced synchronization order parameter, *Λ*; (**d**) Evoked synchronization, *Λ*. Dashed lines = prestimulus responses; solid lines = poststimulus responses. Bottom abscissa scale indicates time points for simulated poststimulus responses; top abscissa scale indicates time points for simulated prestimulus responses. GFP values are in *μV*, order parameters are dimensionless.

**Table 1 entropy-21-00061-t001:** Mean observed induced EEG integration and complexity.

		Observed Data		Surrogate Data	
Task	Condition	I(X)	C_I_(X)	I(X)	C_I_(X)
*1-Exemplar Task*	*Prestimulus*	214.59 (1.07)	258.41 (0.83)	102.54 (100.00, 105.09)	239.95 (237.65, 242.25)
	*Poststimulus*	219.71 (1.39)	254.24 (0.94)	101.74 (99.23, 104.24)	240.99 (238.30, 243.68)
*2-Exemplar Task*	*Prestimulus*	214.21 (1.09)	258.49 (0.78)	102.72 (100.02, 105.42)	239.66 (237.31, 242.00)
	*Poststimulus*	218.66 (1.30)	254.65 (0.88)	101.28 (98.90, 103.66)	240.26 (237.63, 242.88)
*Resting Task*	*Eyes Open*	228.82 (1.20)	242.49 (0.89)	106.29 (103.56,109.01)	244.89 (242.51,247.27)
	*Eyes Closed*	232.22 (1.08)	240.73 (0.83)	101.74 (107.90,113.88)	244.82 (242.45,247.18)

Note: All values are in bits; SE in parentheses for observed data, 95% CIs in parentheses for surrogate data.

**Table 2 entropy-21-00061-t002:** Analysis of variance (ANOVA) results for induced EEG integration and complexity.

Task	EEG Measure	Effect	F	P	ε	η^2^_P_
*Categorization*	I(X)	Task	1.58	0.228	–	0.10
		TI	92.59	0.001	–	0.86
		Task x TI	3.3	0.097	–	0.17
	C_I_(X)	Task	0.41	0.532	–	0.03
		TI	190.59	0.001	–	0.93
		Task x TI	2.10	0.168	–	0.12
*Resting State*	I(X)	RS	34.3	0.001	–	0.70
	C_I_(X)	RS	22.55	0.001	–	0.60
*Categorization vs. Resting State*	I(X)	DC	502.99	* 0.001	0.94	0.97
	C_I_(X)	DC	1738.45	* 0.001	0.83	0.99

ANOVA factor labels: Task, Behavioral Task; TI, Time Interval; RS, Resting State; DC, Data Condition. DC factor effects dfs = 3, 45; all other dfs = 1, 15. Asterisks indicate *p*-values subject to Greenhouse-Geisser correction (see Section 2.8.1).

**Table 3 entropy-21-00061-t003:** Mean observed categorization task evoked EEG integration and complexity.

		Observed Data		Surrogate Data	
Task	Condition	I(X)	C_I_(X)	I(X)	C_I_(X)
*1-Exemplar Task*	*Prestimulus*	213.56 (1.81)	259.75 (1.47)	103.01 (100.48, 105.55)	233.07 (229.53, 236.60)
	*Poststimulus*	242.20 (3.11)	241.34 (2.36)	110.73 (107.68, 113.78)	232.90 (229.71, 236.09)
*2-Exemplar Task*	*Prestimulus*	212.49 (1.45)	258.67 (1.59)	102.73 (100.08, 105.38)	233.34 (230.14, 236.54)
	*Poststimulus*	246.66 (1.83)	238.09 (2.44)	112.05 (108.75, 115.36)	233.06 (228.83, 237.29)

Note: All values are in bits; SE in parentheses for observed data, 95% CIs in parentheses for surrogate data.

**Table 4 entropy-21-00061-t004:** Analysis of variance (ANOVA) results for evoked and induced vs. evoked EEG integration and complexity.

EEG Type	EEG Measure	Effect	F	P	η^2^_P_
*Evoked*	I(X)	Task	2.91	0.109	0.16
		TI	282.84	0.001	0.95
		Task x TI	9.04	0.009	0.38
	C_I_(X)	Task	2.90	0.109	0.16
		TI	64.49	0.001	0.81
		Task x TI	0.91	0.356	0.06
*Induced vs.* *Evoked*		EEG	36.74	0.001	0.71
	I(X)	TI	427.41	0.001	0.97
		EEG x TI	162.93	0.001	0.92
		EEG	21.56	0.001	0.59
	C_I_(X)	TI	103.38	0.001	0.87
		EEG x TI	36.30	0.001	0.71

ANOVA factor labels: Task, Behavioral Task; TI, Time Interval; EEG, EEG Type. All dfs = 1, 15.

**Table 5 entropy-21-00061-t005:** Observed EEG global field power (GFP) and synchronization (Λ) averaged across electrode and time.

Task	Data Condition	GFP_Induced_	GFP_Evoked_	Λ_Induced_	Λ_Evoked_
*1-Exemplar Task*	*Prestimulus*	16.82 (1.48)	3.95 (0.36)	0.209 (0.006)	0.208 (0.015)
	*Poststimulus*	15.83 (1.38)	4.09 (0.37)	0.199 (0.006)	0.248 (0.015)
*2-Exemplar Task*	*Prestimulus*	17.51 (1.68)	4.37 (0.36)	0.218 (0.007)	0.224 (0.010)
	*Poststimulus*	15.97 (1.49)	4.47 (0.37)	0.201 (0.006)	0.243 (0.016)
*Resting Task*	*Eyes Open*	23.50 (2.07)	–	0.202 (0.007)	–
	*Eyes Closed*	20.88 (1.79)	–	0.223 (0.010)	–

GFP values are in *μV*, order parameters are dimensionless; SE in parentheses.

**Table 6 entropy-21-00061-t006:** Analysis of variance (ANOVA) results for induced EEG power and synchronization.

Task	EEG Measure	Effect	F	P	ε	η^2^_P_
*Categorization*	GFP_Induced_	Task	0.46	0.508	–	0.03
		TI	41.41	0.001	–	0.73
		Task x TI	7.02	0.018	–	0.32
	Λ_Induced_	Task	3.16	0.096	–	0.17
		TI	31.05	0.001	–	0.67
		Task x TI	6.80	0.020	–	0.31
*Resting State*	GFP_Induced_	RS	33.42	0.001	–	0.69
	Λ_Induced_	RS	6.79	0.020	–	0.31
*Categorization* vs. *Resting State*	GFP_Induced_	DC	29.94	* 0.001	0.35	0.67
	Λ_Induced_	DC	6.47	* 0.010	0.49	0.30

ANOVA factor labels: Task, Behavioral Task; TI, Time Interval; RS, Resting State; DC, Data Condition. DC factor effects dfs = 3, 45; all other dfs = 1, 15. Asterisks indicate *p*-values subject to Greenhouse-Geisser correction (see Section 2.8.1).

**Table 7 entropy-21-00061-t007:** Analysis of variance (ANOVA) results for evoked and induced vs. evoked EEG power and synchronization.

EEG Type	EEG Measure	Effect	F	P	η^2^_P_
*Evoked*	GFP_Evoked_	Task	4.98	0.041	0.25
		TI	8.14	0.012	0.35
		Task x TI	1.15	0.300	0.07
	Λ_Evoked_	Task	0.83	0.378	0.05
		TI	5.98	0.027	0.29
		Task x TI	0.94	0.349	0.06
*Induced* vs. *Evoked*		EEG	72.47	0.001	0.83
	GFP_Evoked_	TI	33.71	0.001	0.69
		EEG x TI	45.87	0.001	0.75
		EEG	5.63	0.031	0.27
	Λ_Evoked_	TI	1.36	0.262	0.08
		EEG x TI	14.76	0.002	0.50

ANOVA factor labels: Task, Behavioral Task; TI, Time Interval; EEG, EEG Type. All dfs = 1, 15.

**Table 8 entropy-21-00061-t008:** Generalized estimating equation (GEE) regressions relating EEG power and synchronization to EEG integration and complexity.

IV	DV	β	Wald χ^2^	P
GFP_Induced_	I(X)	0.56 (0.11)	27.90	0.004
	CI(X)	−0.52 (0.11)	21.28	0.004
Λ_Induced_	I(X)	−0.27 (0.08)	11.12	0.004
	CI(X)	0.33 (0.07)	22.18	0.004
GFP_Evoked_	I(X)	0.28 (0.04)	40.44	0.004
	CI(X)	−0.29 (0.08)	14.18	0.004
Λ_Evoked_	I(X)	−0.001 (0.11)	0.00	0.965
	CI(X)	0.08 (0.13)	0.36	0.546

All dfs = 1, 16. β-value SEs in parentheses. P-values have been corrected for multiple comparisons across each set of tests separately for each DV (see Section 2.8.1).

**Table 9 entropy-21-00061-t009:** Mean simulated categorization task EEG properties.

Data Condition	Prestimulus	Poststimulus	F	p	η^2^_P_
*I(X)_Induced_*	221.08 (0.09)	237.25 (0.20)	7007.88	0.001	0.99
*C_I_(X)_Induced_*	253.43 (0.16)	244.24 (0.18)	1849.51	0.001	0.99
*I(X)_Evoked_*	221.50 (1.08)	246.78 (1.54)	313.07	0.001	0.95
*C_I_(X)_Evoked_*	255.28 (1.05)	240.36 (1.81)	48.87	0.001	0.77
*GFP_Induced_*	30.63 (0.15)	27.83 (0.09)	310.58	0.001	0.95
*GFP_Evoked_*	9.40 (0.37)	9.64 (0.39)	33.03	0.001	0.69
Λ*_Induced_*	0.195 (0.001)	0.152 (0.001)	1996.46	0.001	0.99
Λ*_Evoked_*	0.173 (0.012)	0.225 (0.007)	16.55	0.001	0.53

Complexity and integration values are in bits, GFP values are in μV, and order parameters are dimensionless; SE are in parentheses. ANOVA parameters describe the significance of prestimulus versus poststimulus differences.

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
