# Peer review of "K-th Nearest Neighbor (KNN) Entropy Estimates of Complexity and Integration from Ongoing and Stimulus-Evoked Electroencephalographic (EEG) Recordings of the Human Brain"

_entropy, 2019, doi:10.3390/e21010061_

Round 1

Reviewer 1 Report

Summary:

The author applied two information theoretic measures, integration I(X) and interaction complexity C(X), to evoked and induced EEG during visual tasks and EEG of resting state. Considering the non-stationarity of induced and evoked EEGs he used non-parametric k-th nearest neighbor (KNN) entropy estimation arguing that KNN entropy estimation is better than the Gaussian-based entropy estimation. He found that I(X) and C(X) differentiate induced and evoked EEG signals and suggested that the increased I (X) and decreased C (X) imply that the brain state under visual perception deviates from the critical state of the brain. In addition, the author simulated the alpha band EEG properties of categorization task with a dipole model, and also applied surrogate data test to assess the spurious I(X) and C(X) from volume conduction and evaluated the significances of changed I(X) and C(X). He released the relevant MATLAB codes which improve the reliability of this study.

This study includes the experiment of visual task, modeling, an improved entropy estimation method, and surrogate data test. The paper was well written and provided proper interpretations of the analysis results and discussed the limitations carefully. Although the good readability of this paper, I have a couple of minor technical questions. If the author answers the questions, that would improve the paper.

Minor technical questions:

Why did the author focus on the alpha band, not theta and gamma band? Does the visual perception associate only with the alpha band? How does the result of integration and interaction complexity of global alpha waves associate with the function of the alpha band previously studied and furthermore how do the results of I(X) and C(X) advance the understanding on the function of alpha waves during visual perception?

The KNN is not a fully nonparametric measure because of the hidden parameters such as K=16. The induced, evoked, and resting state EEG might have significantly different signal properties (i.e., variance), thus each data set might have different K. How did the author test and determine whether k=16 is appropriate for the all datasets?

Because of the uncertainty whether or not the hidden parameters (k, d) are appropriate for each data set, it is also uncertain whether the KNN entropy estimation is better than the Gaussian-based entropy estimation for the given datasets (induced, evoked, and resting state). The author emphasized several time and assumed for the outperform of the KNN entropy estimation, but I am not sure how the author can guarantee its outperform without a quantitative test and a parameter search for the given datasets.  

Author Response

Reviewer Concern #1: “Why did the author focus on the alpha band, not theta and gamma band? Does the visual perception associate only with the alpha band? How does the result of integration and interaction complexity of global alpha waves associate with the function of the alpha band previously studied and furthermore how do the results of I(X) and C(X) advance the understanding on the function of alpha waves during visual perception?”

Author Reply: I thank the reviewer for these very good questions. I described my reasons for focusing on the alpha band on p. 4, lines 142 – 147 of the original manuscript (same page and line numbers in the revised manuscript). I focused on band-filtered signals because such filtering removes data characteristics (e.g. trends, spikes) that reflect nonstationarities other than variance-nonstationarity, a fact consistent with the suggestion that EEG integration and complexity computations will be more accurate when performed on data within a narrow frequency range. I focused on the alpha range in particular because there is evidence that visual-evoked EEG signals have significant energy in the alpha range that is related to visual perception.

That said, the reviewer is correct that the theta and beta frequency bands do correlate with visual perception. (The relationship of the gamma band to perception is controversial because it is difficult to distinguish brain-originating gamma activity with gamma-band noise due to muscle activity. Also, true gamma activity appears to reflect local brain activity rather than the activity of distributed brain networks, and thus should be less reflective of statistical dependencies between EEG sensors.) Hence in the in the discussion section of the revised manuscript (pp. 28 – 29, lines 1058 – 1064), I now state that the focus on alpha band activity is a limitation of the present study. Also, I now include a brief discussion of the functional meaning of alpha-range activity with respect to visual perception and how the present EEG integration and complexity results advance our understanding of this issue (see pp. 25, lines 866 – 882). I should also note that both the original and revised versions of the manuscript also include(d) a discussion of what the present findings mean for our more general understanding of brain dynamics in terms f the phenomenon of criticality (p. 23, lines 838 – 858 of the original manuscript; p. 26, lines 926 – 946 of the revised manuscript). 

Reviewer Concern #2: “The KNN is not a fully nonparametric measure because of the hidden parameters such as K=16. The induced, evoked, and resting state EEG might have significantly different signal properties (i.e., variance), thus each data set might have different K. How did the author test and determine whether k=16 is appropriate for the all datasets?”

“Because of the uncertainty whether or not the hidden parameters (k, d) are appropriate for each data set, it is also uncertain whether the KNN entropy estimation is better than the Gaussian-based entropy estimation for the given datasets (induced, evoked, and resting state). The author emphasized several time and assumed for the outperform of the KNN entropy estimation, but I am not sure how the author can guarantee its outperform without a quantitative test and a parameter search for the given datasets.” 

 Author Reply: The author is correct that the KNN is parametric in the sense that it depends on the parameter k. In the submitted manuscript, the term “nonparametric” refers to the conventional statistical meaning of “no distributional assumptions” (typically assumptions of normally-distributed data). The value of this parameter used in the present study was k = 36 (not 16); I described my reasons for this choice in detail on p. 8, lines 301-313 of the original manuscript (pp. 8-9, lines 303-315  in the revised manuscript). In brief, the value of the nearest neighbor parameter k should be small in order to provide a local estimate of the geometry of the joint variable data space, but it should also be at least equal to the number of dimensions of the data space in order to correctly compute the local ellipsoidal neighborhood around each data point by SVD for the G-KNN entropy estimate (see Lord et al., 2018, Chaos, 28: 033114). Given that the data space analyzed via the KNN measure had 36 dimensions, the choice of k = 36 met both of these criteria (i.e. it was the smallest value of k that was equal to or larger than the dimension of the data space). These two criteria (small k, but larger than the data space dimension) are general criteria that are applicable to and appropriate for any data set.

I did not conduct extensive testing using larger values of k because 1) it would violate the first of these two necessary criteria, 2) it was computationally-intractable to do so, and 3) because larger values of k produce less accurate entropy estimates (although the variance of such estimates is reduced with larger k; see Gao et al., 2014, arXiv 2014, 1411.2003; Khan et al. 2007. Physical Review E, 76: 026209). That said, even though the choice of k was based on theoretical principles, I think it is fair to consider the use of a single value of k as a limitation of the present study that could be explored in future research. I now discuss this issue in more detail as a study limitation in the revised manuscript (see p. 28, lines 1030 – 1046), but I do not think this issue is relevant to the issue of whether the use of the KNN-based or Gaussian-based estimator is more appropriate. I think this because of the results of additional analyses I now report in the revised manuscript (see below).

With concern to the variable “d’ in Equations 6 and 7, I respectfully disagree with the reviewer that this variable is a free parameter like k. The variable d simply indicates the size of the data space, i.e. the number of variables to be related to each other; this is presumably given at the beginning of an analysis and thus is not necessarily “free” in the sense that it can be set to an arbitrary value in order to produce the best estimate of the data entropy. That said, for the present analysis I did reduce the initial data space from d = 72 to d = 36 (I reduced the number of time points entering the analysis as well) in order to reduce the computation time. I reported how I did this on pp.7-8, lines 259 – 270 of the original manuscript (p. 8, lines 267 – 272 of the revised manuscript) and I discussed my rationale for this on pp. 23-24, lines 874 – 890 of the original manuscript (p. 27, lines 977 – 1000 of the revised manuscript). In brief, the data downsampling was performed to make the necessary surrogate data creation computationally-tractable, and I make explicitly clear that these computational difficulties are a drawback of the KNN method for large variable spaces. However, I do not think this is a limitation of the present study because in this discussion I also state that “initial exploration of the downsampling procedure did not alter the pattern of I(X) and CI(X) observed without the downsampling; hence the results from the downsampled data were reported in this paper in order to maintain consistency across surrogate and non-surrogate analyses”. The fact that the same basic pattern of I(X) and CI(X) was found for d = 36 and d = 72 (the maximum possible value for this data space) shows that the choice of this parameter (like parameter k) is also not relevant to the issue of whether or not the use of the KNN-based or Gaussian-based estimator is more appropriate.

In the revised manuscript, I now address the KNN-based versus Gaussian-based estimation issue more directly by including an analysis where the EEG data are first transformed to follow a normal distribution following my previous work (Trujillo et al., 2017, Frontiers in Neuroscience, 11, 425) and then I(X) and CI(X) are computed by 1) KNN-based estimation and 2) Gaussian-based estimation. I also verified that, prior to the Gaussian transformation, the EEG data significantly deviated from normality for all subjects; see revised Supplementary Material Section 1.3. (Although interestingly, EEG integration and complexity values computed from non-normal data are normally-distributed across subjects; see my reply to Reviewer #2 for further detail). For both the categorization task EEG data, I found different I(X) and CI(X) patterns for the non-normal versus normal data when using the KNN estimator and the Gaussian-based estimator. In the case of the resting state EEG data, I found different I(X) and CI(X) patterns for the non-normal versus normal data when using the KNN estimator, but the same I(X) and CI(X) patterns for the non-normal versus normal data when using the Gaussian-based estimator.

This analysis confirms a main conclusion of the manuscript that the KNN estimator should be utilized in most cases when the data distribution is unknown or non-normal, and when the data is nonstationary. This analysis shows that, in some cases, the particular form of the distribution of a data set determines the outcome of the entropy estimation. Normal-data produces one pattern of results via the KNN estimator, whereas non-normal data produces a different pattern of results for the same KNN estimator. This then argues that transformation of the data distribution to Gaussian and then using the Gaussian-based estimator should be avoided in most cases. This is because such data transformation changes the distribution of the data away from its true distribution and thus true entropy values. This may have been the case for the categorization task EEG data because such signals are inherently nonstationary. The only time such data transformations should be used is when it can be shown that it doesn’t make much of a difference. This was the case for the resting state data because the same basic pattern of results obtains when applying a Gaussian-transformation and then a Gaussian entropy estimator, or when applying the KNN estimator to non-transformed resting state data.

Please note that I report the general finding of this analysis in the main text of the revised manuscript (see Section 3.5), but I report the detailed quantitative results in the revised Supplemental Materials (see Sections 1.4). I do this because the additional tables and text needed to fully report this analysis would make the main text much longer and more difficult to read. However, I can place the full details of this analysis in the main text if the reviewer(s) and the editors feel it would be more appropriate.

Reviewer 2 Report

The article introduces two Information-theoretic measures estimated by K Nearest Neighbor (KNN) entropy to discriminate between induced and evoked electroencephalographic (EEG) signals from visual stimuli. The study is clearly structured and described, and the methodology proposed opens interesting perspectives for EEG-based exams in clinics. I have a few comments:

-        How was bad channel interpolation performed? How would the results change if channels were completely removed from the analysis?

-        Section 2.6, Page 8, Line 311: please replace “ne cessary” with “necessary”

-        ANOVA analysis can be applied only to normally distributed data. Did the Author check data distribution and performed suitable statistical tests accordingly?

-        Table 1: please reformat the table to include the p-value for the comparison Prestimulus vs Poststimulus and Eyes Open vs Eyes Closed.

-        Similarly for other statistical tests, try to incorporate at least a few parameters (p values, Wald indices, etc.) in the tables to make the manuscript more readable.

-        It may be interesting to introduce a comparison with other approaches (e.g. Gaussian-based entropy) to corroborate the discriminant accuracy of the proposed features.

Author Response

Reviewer Concern #1: “How was bad channel interpolation performed? How would the results change if channels were completely removed from the analysis?”

Author Reply: Bad channel interpolation was performed using a standard EEGLAB-based spherical spline interpolation algorithm [Perrin, Perrier, Bertrand, Giard, & Echallier, 1987; m=5; 50-term expansion], which is fairly robust. The effect of such interpolation might be to increase the statistical dependency of the interpolated signal(s) with the other EEG signals. However, as the interpolations were applied to all data conditions, any interpolation-related dependency increase should be subtracted out during the between-condition contrasts performed in this study, and contribute to the creation of the surrogate statistical distributions (i.e. such dependencies would serve to make the surrogate tests more conservative).  Moreover, the mean number of interpolated channels was only 2.8 ± 0.9 across subjects. As this is less than 4% of channels, it is unlikely that the results would change at all if the bad channels were removed from the analysis. Also, any effect of interpolation on the final analysis would be further reduced given the fact that neighboring pairs of channels were averaged together during the spatial downsampling step (see Section 2.4 of the revised manuscript), which in itself is a form of spatial interpolation. Given all of the above, and the fact that removing channels from the analysis could raise other difficulties (such as the creation of an unbalanced inter-electrode dependency structure across the scalp due to the idiosyncratic locations of the interpolated channels across subjects), I have chosen to retain analysis incorporating the interpolated channels in the revised manuscript. However, I now report the interpolation algorithm used and the average number of interpolated channels in the revised manuscript (see Section 2.4, lines 246-249).

Reviewer Concern #2: “Section 2.6, Page 8, Line 311: please replace “ne cessary” with “necessary””.

Author Reply: Thank you for pointing out this typographic error. The requested correction has been performed.

Reviewer Concern #3: “ANOVA analysis can be applied only to normally distributed data. Did the Author check data distribution and performed suitable statistical tests accordingly?”

Author Reply: I did not initially check this, but in response to this concern, I performed Jarque-Bera tests of univariate normality for each EEG and behavioral measure across subjects. Given the large number of tests, Type-I error was minimized by correcting the p-values for multiple comparisons using the Holm-Bonferroni procedure. None of the EEG measures were significantly different from a normal distribution (ps > 0.14). I should note, however, that even though the final EEG integration and complexity values entering into the ANOVA were normally-distributed across subjects, the EEG data from which they were computed were non-normal (see my reply to Reviewer #1 for further detail). This suggests that the operations involved in the computation of I(X) and CI(X) smooth the data in accordance with the Central Limit Theorem. Any concern that this approach was overly conservative and thus might lead to Type-II error in this analysis (i.e. some of the measures were non-normal, but did not depart from normality enough to yield a significant test ) may be mitigated by the fact that ANOVAs and GEEs are fairly robust to minor violations of distributional assumptions (Hubbard et al., 2010, Epidemiology, 21, 467-474; Glass et al.,1972. Rev. Educ. Res. 42, 237-288; Harwell et al. 1992. J. Educ. Stat. 17, 315-339; Lix et al.,1996. Rev. Educ. Res. 66, 579-619). Thus I have retained the use of ANOVAs and GEE-based regression for these measures in the revised manuscript and I have also reported the results of this distribution check in the revised Supplemental Materials (Section 1.2).

            The distribution of the behavioral reaction time data was also not significantly different from normal for both the 1-Exemplar and 2-Exemplar Categorization tasks. However, while the 2-Exemplar task accuracy rates were normally-distributed, the 1-Exemplar accuracy rates were not; this is because accuracy in this task was near ceiling (92% - 100%). I accounted for this in two ways in the revised manuscript. First, I used a nonparametric Wilcoxon signed rank test in order to assess accuracy differences between the two categorization tasks. Second, for GEE-based regressions relating accuracy to EEG integration and complexity, I treated accuracy as the independent variable and the EEG measures as the dependent variable; DVs are model-dependent in GEE analyses, whereas IVs are not (see Hubbard et al., 2010, Epidemiology, 21, 467-474). Moreover, the GEE analyses used a robust covariance estimator, which was also model-independent. I have reported the above rationale in Section 1.2 of the revised Supplemental Materials.

Reviewer Concern #4:  “Table 1: please reformat the table to include the p-value for the comparison Prestimulus vs Poststimulus and Eyes Open vs Eyes Closed.”

“Similarly for other statistical tests, try to incorporate at least a few parameters (p values, Wald indices, etc.) in the tables to make the manuscript more readable.”

Author Reply: I agree that the large number of statistical tests performed in this study can be difficult to assess when reported as text. However, I did not reformat Table 1 as requested because there was not enough space in the table to include such information, while keeping the text font legible. Instead, in the revised reporting of the statistical tests of the EEG measures, I have reported the omnibus ANOVA and GEE results in separate tables (there are now 5 additional tables). However, I have retained reporting of the statistical tests of behavior in the text because the number of such tests was relatively small and the addition of a 6th table would have made the revised manuscript longer and more cumbersome to read. I hope the reviewers and editors find my solution to this issue to be satisfactory.

Reviewer Concern #5:  “It may be interesting to introduce a comparison with other approaches (e.g. Gaussian-based entropy) to corroborate the discriminant accuracy of the proposed features.”

Author Reply: In the revised manuscript, I now address the KNN-based versus Gaussian-based estimation issue more directly by including an analysis where the EEG data are first transformed to follow a normal distribution following my previous work (Trujillo et al., 2017, Frontiers in Neuroscience, 11, 425) and then I(X) and CI(X) are computed by 1) KNN-based estimation and 2) Gaussian-based estimation. I also verified that, prior to the Gaussian transformation, the EEG data significantly deviated from normality for all subjects; see revised Supplementary Material Section 1.3. (Although interestingly, EEG integration and complexity values computed from non-normal data are normally-distributed across subjects; see my reply to Reviewer #2 for further detail). For both the categorization task EEG data, I found different I(X) and CI(X) patterns for the non-normal versus normal data when using the KNN estimator and the Gaussian-based estimator. In the case of the resting state EEG data, I found different I(X) and CI(X) patterns for the non-normal versus normal data when using the KNN estimator, but the same I(X) and CI(X) patterns for the non-normal versus normal data when using the Gaussian-based estimator.

This analysis confirms a main conclusion of the manuscript that the KNN estimator should be utilized in most cases when the data distribution is unknown or non-normal, and when the data is nonstationary. This analysis shows that, in some cases, the particular form of the distribution of a data set determines the outcome of the entropy estimation. Normal-data produces one pattern of results via the KNN estimator, whereas non-normal data produces a different pattern of results for the same KNN estimator. This then argues that transformation of the data distribution to Gaussian and then using the Gaussian-based estimator should be avoided in most cases. This is because such data transformation changes the distribution of the data away from its true distribution and thus true entropy values. This may have been the case for the categorization task EEG data because such signals are inherently nonstationary. The only time such data transformations should be used is when it can be shown that it doesn’t make much of a difference. This was the case for the resting state data because the same basic pattern of results obtains when applying a Gaussian-transformation and then a Gaussian entropy estimator, or when applying the KNN estimator to non-transformed resting state data.

Please note that I report the general finding of this analysis in the main text of the revised manuscript (see Section 3.5), but I report the analysis results in the revised Supplemental Materials (see Sections 1.4). I do this because the additional tables and text needed to fully report this analysis would make the main text much longer and more difficult to read. However, I can place the full details of this analysis in the main text if the reviewer(s) and the editors feel it would be more appropriate.